# Finely-resolved along-track wave attenuation estimates in the Antarctic marginal ice zone from ICESat-2

Joey J. Voermans[1], Alexander D. Fraser[2,3], Jill Brouwer[2,3,*], Michael H. Meylan[4], Qingxiang Liu[5,1], and Alexander V. Babanin[1]

[1]Department of Infrastructure Engineering, University of Melbourne, Parkville, Victoria, Australia
[2]Australian Antarctic Program Partnership, Institute for Marine and Antarctic Studies, University of Tasmania, nipaluna / Hobart, 7001 Tasmania, Australia
[3]Institute for Marine and Antarctic Studies, University of Tasmania, nipaluna / Hobart, 7001 Tasmania, Australia
[4]School of Information and Physical Sciences, University of Newcastle, Callaghan, 2308 NSW, Australia
[5]Frontier Science Center for Deep Ocean Multispheres and Earth System (FDOMES) and Physical Oceanography Laboratory, Ocean University of China, Qingdao, China
[*]Now at NCMI Engineering and Technology, CSIRO, Australia

**Correspondence:** Joey J. Voermans (jvoermans@unimelb.edu.au)

**Abstract.** Advances in our modeling capacity of wave-ice interactions are hindered by the limited availability of wave observations in sea ice and, specifically, under a broad range of wave and sea ice conditions. Satellite remote sensing provides opportunities to vastly expand the observational dataset of waves in sea ice and the study of wave-ice interactions. Specifically, Brouwer et al. (2022) demonstrated a clear reduction of observed wave energy into the Antarctic Marginal Ice Zone
5 (MIZ) as derived from ICESat-2 observations. Here, we build upon the work of Brouwer et al. (2022) to estimate the wave attenuation rate in the Antarctic MIZ under a wide variety of sea ice conditions. Overall statistics of the observations reveal a linear increase in the wave attenuation rate with relative distance into the MIZ, implying that the wave energy in the MIZ scales as $\sim \exp(\beta x^2...)$, where $\beta$ is a frequency-dependent attenuation coefficient. Attenuation rates are well-sorted with wave frequency, where highest attenuation rates are observed for the shortest waves. We find that both the magnitude and frequency
10 dependence of the ICESat-2 estimated wave attenuation rates are consistent with in situ observations. We further highlight that the misalignment between the incident wave direction and the measurement transect, and the inhomogeneity of the ice pack may lead to significant local fluctuations and negative values in the estimated wave attenuation rate when evaluating individual transects. The strong dependence of the overall statistics of the wave attenuation rate on the wave frequency and the relative distance into the MIZ alone provides significant opportunities in modeling wave-ice interactions in the Antarctic environment
15 at global and climate scales, as it does not depend on system variables that are not straightforward to measure, retrieve or simulate at such large scales. However, independent parameterization of the MIZ width will be required to do so.

## 1 Introduction

The Antarctic Marginal Ice Zone (MIZ), the region separating the Southern Ocean from the consolidated pack ice, represents
an interface of intense air-sea-ice interactions (e.g., Häkkinen, 1986; Weeks, 2010; Squire, 2020; Bennetts et al., 2024). Ocean
waves are a critical mediator between the ocean, ice and atmosphere in the MIZ due to their capacity to break the ice (e.g.,
Kohout and Meylan, 2008; Dumont et al., 2011; Voermans et al., 2020), and, in doing so, waves can rapidly alter the sea ice
morphology, its mobility, and, as a consequence, modulate the air-sea-ice fluxes in the MIZ (Williams et al., 2013; Collins III
et al., 2015; Boutin et al., 2020; Li et al., 2021) such as the lateral melt of floes and thus the seasonal cycle during ice meltback
(Bateson et al., 2022).

The spatial extent to which waves can impact the ice cover, and thus the associated MIZ dynamics, is in large part determined
by the amount of wave energy that propagates through the ice, which can, under certain conditions, propagate over a thousand
kilometers into the ice pack (Nose et al., 2023; Squire et al., 2009). Wave energy attenuates through various scattering and
wave dissipation mechanisms which largely depend on the properties of the sea ice and wave field, including sea ice thickness,
concentration, floe size distribution, sea ice mechanical and material properties, wave frequency and wave steepness (for and
overview see Shen, 2019; Squire, 2020; Rogers et al., 2021; Thomson, 2022, and references therein). To model waves in sea ice,
a considerable number of theories and parameterizations have been established to estimate the wave attenuation rate (e.g., Liu
et al., 2020, their Table 1). While the majority of these have been calibrated and/or validated using experimental observations,
uncertainty persists under what sea ice and wave conditions the various physical processes that underpin these models may be
valid or of importance (Shen, 2019; Voermans et al., 2021; Shen, 2022).

Our understanding and capacity to model the wave attenuation rate in terms of system variables is, in part, hindered by the
characteristic length scales associated with the wave-ice interactions mechanisms, the numerical models and sea ice observa-
tions. Operational forecasting models and wave observations typically operate or are retrieved at scales of $O(1-10 \, \text{km})$. Yet,
sea ice conditions and wave-ice physics can vary strongly across such distances, where, for example, individual floe dimensions
can vary from $O(\text{m})$ to $O(\text{km})$ (Toyota et al., 2006), and ice-ocean roughness scales can vary from $O(\text{mm})$ for smooth surfaces
to $O(\text{m})$ for ridges. In most cases, MIZ consists of a mixture of different types of ice and open water conditions where each
combination may lead to completely different behaviors of waves in ice (Herman, 2024). This not only makes it challenging
as to how the dynamics at small scales need to be captured at much larger scales, whether it be in the form of an effective or
average sea ice property, but also how such properties can be realistically derived or observed at much larger scales for usage
in numerical simulations. In particular, products of only sea ice concentration and sea ice thickness are currently routinely
generated in the Antarctic. Given the many variables required to describe sea ice, this often leads to the necessary assumption
that all other sea ice variables in our models are homogeneous, ultimately representing a major source of uncertainty.

A straightforward approach to improve our modeling capacity of wave-ice interactions is to retrieve considerably more data
under a diverse range of wave and sea ice conditions in an attempt to identify trends with system variables (Rogers et al., 2021;
Montiel et al., 2022; Rabault et al., 2023). Observations of the wave attenuation rate in sea ice can be obtained by measuring
the difference in wave energy between two locations, typically using wave-ice buoys deployed on the ice or by any other

motion-recording sensor. Traditionally, it is assumed that wave energy decays exponentially with distance into the MIZ (e.g., Wadhams et al. (1988), although questions have been raised on the a priori assumption of its validity; e.g., see Squire (2018); Herman (2024)):

$$E(f,x) = E(f,0)\exp(-\alpha x) \tag{1}$$

where $E(f,x)$ is the wave energy spectrum, which varies with wave frequency $f$ and the distance into the MIZ $x$ at a rate given by the wave attenuation rate $\alpha$. The number of estimates of the wave attenuation rate from wave observations has increased drastically in recent years due the rapid progresses in the development of low-cost wave-ice buoys, as they are able to capture the wave field characteristics $E(f,x)$ at high temporal resolution and at high accuracy (Kohout et al., 2015; Rabault et al., 2022; Kodaira et al., 2024; Womack et al., 2024). The main restriction of in-situ deployments is, however, that the spatial coverage tends to be extremely sparse due to the logistical challenges in deploying instrumentation in the remote and harsh Antarctic MIZ. Thus, even though the studies associated with such deployments may provide high temporal detail of wave-ice interactions, the complexity and diversity of sea ice conditions during such deployments restricts our general understanding of wave-ice interactions in general and their applicability at much larger scales.

Satellite remote sensing may provide critical large spatial coverage that is difficult to achieve with in-situ instruments alone, albeit, at the cost of reduced temporal and frequency resolution. Synthetic Aperture Radar (SAR) imagery was used by Ardhuin et al. (2017) to retrieve wave field properties in the MIZ, and by Stopa et al. (2018) to derive wave attenuation estimates. Horvat et al. (2020) and Collard et al. (2022) used laser altimeter observations from ICESat-2 to identify waves in sea ice, and was used by Brouwer et al. (2022) and Hell and Horvat (2024) to derive 1D and 2D wave spectra, respectively. While no direct estimates of the wave attenuation rate $\alpha$ were provided, the ICESat-2 derived data of Brouwer et al. (2022) showed a clear reduction of the spectral wave energy into the Antarctic MIZ (i.e. see Figs. B2 and C2 in Brouwer et al., 2022). This highlights the potential of using ICESat-2 observations to estimate the wave attenuation rate in sea ice across large distances. In this study, our objective is to estimate the wave attenuation rate from ICESat-2 altimeter measurements in an attempt to identify trends in the wave attenuation rate across the Antarctic MIZ and under a wide range of sea ice conditions.

## 2 Methods

For the estimation of the wave attenuation rate in the MIZ we make use of the processed ICESat-2 data (Fraser et al., 2024) as derived by Brouwer et al. (2022), which is derived from the Level 3 sea ice height product (ATL07, version 2; Kwok et al. (2021)), from the National Snow and Ice Data Center (NSIDC; https://nsidc.org/data/atl07, last access: 5 March 2020). The transects included in this dataset were subjected to extensive quality control measures, including the selection of transects with low cloud coverage and manual discrimination between swell-dominated and ice-structure-dominated contributions to the observed height. The data include 320 transects covering February, May, September and December of 2019, representing times of minimum extent, rapid autumn advance, maximum extent and rapid summer retreat, respectively (Eayrs et al., 2019). The data consist of surface height measurements of the three high-power laser beams crossing the Antarctic MIZ in a predominantly

north-south direction, resampled to a regular horizontal spacing of 8 m (i.e., oversampling the ATL07 mean segment length of ~15 m). The dataset of Fraser et al. (2024) includes estimates of the MIZ width $x_{MIZ}$ and will be used here as well. Following Brouwer et al. (2022), the MIZ width was conceptually defined as the depth of wave penetration into the MIZ: *'where significant wave height attenuation equals the estimated error in significant wave height'*. Here, we use the median value of the MIZ width determined in four wavelengths (165, 239, 345 and 498 m) as proposed by Brouwer et al. (2022). In physical terms, this definition is close to where the variance of the surface elevation transitions from an outer wave dominated region to the inner ice-structure dominated region. The ice edge was taken as the position where the sea ice concentration was 15%. The reader is referred to Brouwer et al. (2022) for further details on the dataset Fraser et al. (2024), the definition of the MIZ width, and initial quality control procedures. For data interpretation purposes, we also use Sentinel-1 imagery, downloaded from EO Browser, https://apps.sentinel-hub.com/eo-browser/, Sinergise Solutions d.o.o., a Planet Labs company (last access: 24 March 2024); AMSR2-derived sea ice concentration (Spreen et al., 2008), downloaded from https://data.seaice.uni-bremen.de/amsr2/ (last access: 22 February 2024), and SMOS sea ice thickness (Huntemann et al., 2014), downloaded from https://data.seaice.uni-bremen.de/smos/ (last access: 22 February 2024), with overall uncertainty of 10% and 10 cm for sea ice concentration and ice thickness, respectively.

Here, the wave energy density $E(k,x)$, where $k$ is the wave number, was estimated for section lengths $L$ along each transect using Welch's method with Hamming windowing, 50% overlap and the section length $L$ was segmented into windows of 128 sample points, i.e., a length of 1024 m. With no information on the wave direction, we assume that the apparent wave number $k_a = 2\pi/\lambda_a$, where $\lambda_a$ is the apparent wave length, is equal to the wave number $k = k_a/\cos(\Delta\theta)$ with $\Delta\theta = 0$. That is, we assume that the wave direction is aligned with the ICESat-2 transect. While this is a common and often necessary assumption in wave-ice interaction studies, this may lead to a systematic bias if the dominant wave direction is not in the north-south direction. Specifically, one may find that $\lambda/\lambda_a = 0.97, 0.87$ and $0.71$ for $\Delta\theta = 15°, 30°$ and $45°$.

The choice of $L$ is somewhat arbitrary. Large $L$ reduces uncertainty in estimates of $E(k,x)$ whilst potential inhomogeneity of the sea ice cover requires a relatively small section length. In Fig. 1(a,b) we compare the wave energy estimates for section lengths $L$ of 2048, 8192 m and 16384 m for two transects. For quality control purposes, we estimate the signal-to-noise ratio (eSNR, where 'e' denotes 'estimated') as $\sqrt{\int_{k_i}^{k_{i+1}} E(k_i,x)\Delta k}/\epsilon$, where we assume an accuracy of the ICESat-2 measurements in sea ice as $\epsilon = 2$ cm, following the study of Neumann et al. (2019). It is noted, however, that this is not a robust estimate of the eSNR as it depends on the resolution of the spectral density estimate $\Delta k$. We use a value of eSNR = 1 as a cut-off threshold. For some transects, the energy in some wave number bins increases again towards the end of the MIZ, which is most likely due to a strong increase of the ice-structure contribution to the surface height fluctuations after most wave energy has been attenuated (as mentioned in Brouwer et al., 2022). For this reason, we choose the absolute minimum of wave energy within the MIZ as a cut-off instead for these transects.

To estimate the apparent attenuation rate $\alpha$, we make use of the commonly adopted assumption that wave energy decays exponentially with distance into the ice pack (Eq. 1). When surface elevation measurements are available at different $x$ (i.e.

from an altimeter transect), $\alpha$ can be readily obtained from:

$$\alpha = \frac{-\ln\left(E(f,x_2)/E(f,x_1)\right)}{\Delta x} \tag{2}$$

with $\Delta x = (x_2 - x_1)\cos(\Delta\theta)$, where $x_1$ being the position closest to the ice edge. While there are different methods for estimating the wave direction, such as from satellite observations (Hell and Horvat, 2024; Collard et al., 2022) or wave model hindcast (Stopa et al., 2016; Liu et al., 2021), we assume here as well that the waves are well aligned with the direction of the ICESat-2 measurements, i.e., $\Delta\theta = 0$, rather than attempting to apply such corrections. We will discuss this assumption later on. Naturally, this means that the attenuation rates estimated in this study are likely an underestimation by a factor of $1/\cos(\Delta\theta)$ which for $\Delta\theta = 15°$, $30°$ and $45°$ corresponds to 1.04, 1.15 and 1.41.

Following Eq. 2, the wave attenuation rates corresponding to the wave energy of transects shown in Fig. 1(a,b) are shown in Fig. 1(c,d) for different $L$ and for $\Delta x = 16$ km. As $L = 8192$ m significantly reduces the fluctuations in $E$ and $\alpha$, we choose $L = 8192$ m as the section length, as opposed to a larger length, to restrict uncertainty associated with sea ice inhomogeneity on our results. Further, in this study we restrict our focus to the wave energy at wave lengths of $\lambda = 128, 227, 341$ and $512$ m, corresponding to wave periods of roughly $T = 9, 12, 15$ and $18$ s in deep water assuming the linear dispersion relationship for open-water waves. We note here that the systematic bias introduced by considering the apparent along-track wavelength rather than the true wave length depends on the misalignment between the wave direction and ICESat-2 track which, considering $\Delta\theta = 15°$, $30°$ and $45°$, is $T/T_a = 0.98, 0.93$ and $0.84$, respectively, where $T$ is the real wave period and $T_a$ the apparent wave period. An overview of the geographical distribution of observations of $\alpha$ for each month after quality control is shown in Fig. 2.

## 3 Results

### 3.1 Attenuation examples

The wave energy and attenuation rate of an example transect are shown in Fig. 3 for spectral bins corresponding to $T = 9, 12, 15$ and $18$ s. A Sentinel-1 image of the sea ice conditions a day after the ICESat-2 measurements is provided in Fig. 3a, showing consolidated pack ice along the majority of the transect, with the exception of the initial part of the MIZ which looks like sea ice with very low concentration. Aside from minor spatial fluctuations in wave energy, an overall decrease of the spectral wave energy is observed from the ice edge into the MIZ for all wave periods (Fig. 3b). The attenuation rate estimated for this transect is well sorted by wave frequency, showing strongest attenuation for the shortest waves (Fig. 3c). The early cutoff of $\alpha$ for $T = 9$ s is a consequence of the high attenuation rate and the eSNR threshold imposed. The attenuation rate for $T = 18$ s declines gently before briefly becoming slightly negative halfway the MIZ. We note that the overall magnitudes of the apparent attenuation rates are comparable to those observed by Kohout et al. (2020), Voermans et al. (2021) and Montiel et al. (2022), and slightly larger than Meylan et al. (2014) and Rogers et al. (2021).

Another example transect is shown in Fig. 4, illustrating contrasting trends in the estimates of the apparent attenuation rates compared to Fig. 3. For this transect, fluctuations and larger scale trends in the estimated wave energy are of sufficient

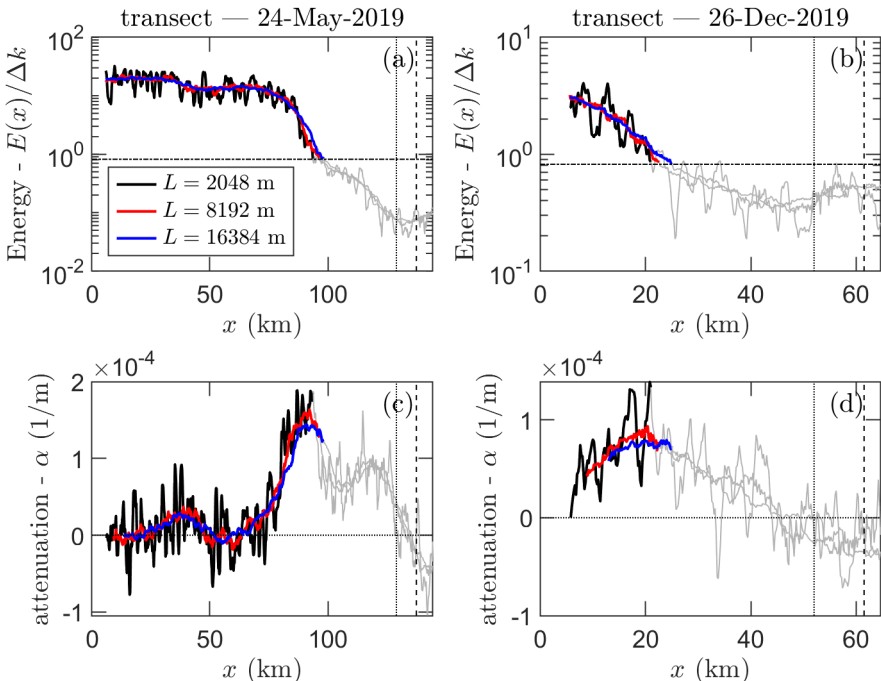

**Figure 1.** Wave energy (a,b) and apparent wave attenuation rates (c,d) estimated for two transects for $T = 12$ s, and for different section lengths $L = 2048$ m, 8192 m and 16384 m. Parts of transect with eSNR $< 1$ are marked in grey. Dotted and dashed vertical lines depict the inner MIZ boundary as determined by Brouwer et al. (2022) corresponding to $T = 12$ s and a broad range of frequencies, respectively.

magnitude to lead to slightly negative estimates of the apparent wave attenuation rate for all wave periods (Fig. 4c). For $x/x_{MIZ} < 0.5$, $\alpha$ does not show any correlation with wave frequency. This changes for $x/x_{MIZ} > 0.5$ where $\alpha$ increases steeply and appears well sorted again with wave frequency. From the Sentinel-1 imagery it is unclear why this sudden change occurred around $x/x_{MIZ} = 0.5$. While a strong gradient in Sentinel-1 backscatter can be observed between $0.1 < x/x_{MIZ} < 0.2$ (Fig. 4a), the location of this gradient does not appear to coincide with the position of rapid change in $\alpha$. Imagery on the days before and after the transect measurements do not suggest great change in location of this strong gradient in backscatter.

### 3.2 Sensitivity of attenuation rate to the wave field direction

While it is tempting to interpret small details of $\alpha$ for each transect in terms of spatial changes in sea ice conditions and/or wave physics, it remains uncertain whether this is realistic. For example, while the occasional negative attenuation rates observed in Fig. 4 would imply an increase in wave energy, potentially due to wind-input (Brenner and Horvat, 2024), non-linear wave interactions (e.g. Li et al., 2017), or wave energy arriving at the observation sites from different directions (e.g., Herman, 2024), they may also simply be methodological artifacts due to a low and/or variable signal-to-noise ratio (Thomson et al., 2021) or non-stationarity of the incoming wave field (Voermans et al., 2023). Here, however, we suggest that some transient fluctuations

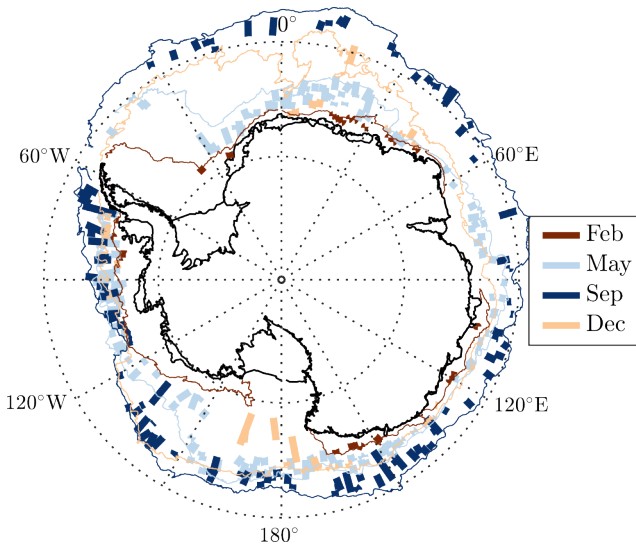

**Figure 2.** Spatial distribution of the wave attenuation rate observations after quality control measures sorted by month. Contours represent an approximation of the ice edge, derived from the monthly average 15% sea ice concentration.

and negative attenuation rates observed in our estimates may be caused by the assumption of homogeneous sea ice combined with the assumption that the waves propagate along the same direction as the measurement transect.

To illustrate this, we take another example transect (Fig. 5a) where sea ice conditions surrounding the ice edge are highly inhomogeneous, most likely due to ocean eddy-ice interactions (e.g., Manucharyan and Thompson, 2017). We then take the Sentinel-1 backscatter intensity as a crude proxy of the local attenuation rate $\alpha \approx \alpha_p$, where we assume that in open water (dark = low backscatter) the attenuation is very low and $O(10^{-6})$, and in ice covered regions (bright = high backscatter) the attenuation rate is much higher, i.e., $O(10^{-4})$ (see colorbar in Fig. 5a). Specifically, we quantify the proxy-attenuation rate $\alpha_p$ to the Sentinel-1 backscatter $\sigma$ based on the distribution of $\sigma$ in the Sentinel-1 backscatter image and typical variability of true $\alpha$ observed in-situ (see Appendix A for further details). We stress that this is by no means an accurate depiction of the true attenuation rate and is merely used here to illustrate the impact of wave direction on the interpretation of the estimated wave energy along a transect.

We then estimate the wave energy along the transect for different wave directions given the spatial variability of $\alpha_p$ and assuming an exponential decay of wave energy with $x$, i.e., Eq. 1. With such a simple model, wave energy traveling in the same direction as the transect (in this case at $6°$ relative to north, we use the 'coming from' convention here) decreases monotonically along the transect (Fig. 5b). However, if waves would approach from either $15°$ or $325°$ relative to north, fluctuations in wave energy along the transect start to appear. If $\Delta x$ is smaller than the length scale of such fluctuations, a negative apparent attenuation rate will be found. In case of $325°$, this leads to a very large variability as waves traveling towards the transect at around $x/x_{MIZ} = 0.4$ experience low attenuation ice conditions when coming from the Southern Ocean in contrast to waves arriving at neighboring points. In other words, waves arriving at different points along the transect are experiencing completely

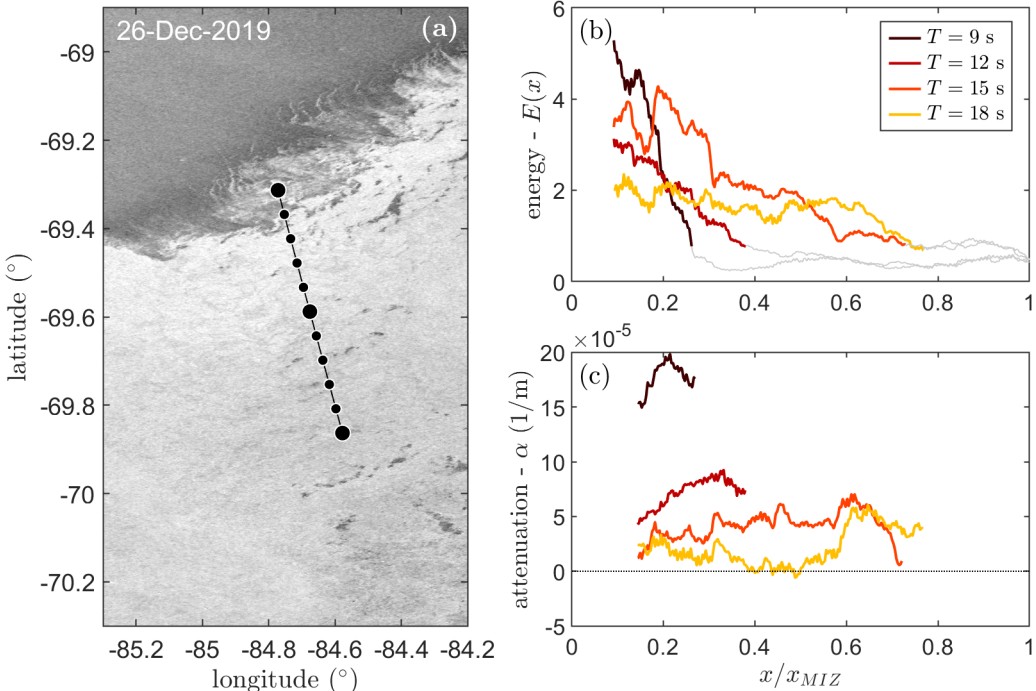

**Figure 3.** An example ICESat-2 transect (a) on 2019-12-26 with estimated spectral wave energy (b) and apparent wave attenuation rate (c). The MIZ, as derived by Brouwer et al. (2022), along the transect is given by the black line (a), which is 61.5 km wide. Black circles denote fractions of the MIZ width, from 0.0 to 1.0, with the larger circles corresponding to $x/x_{MIZ} = 0$, 0.5 and 1. The Sentinel-1 image is from 2019-12-27, i.e., a day after the ICESat-2 transect measurements. The gray lines in (b) represent observations removed after quality control.

different sea ice conditions that lead to completely different magnitudes in the observed wave attenuation. In this specific case, this leads to a threefold increase in the observed wave energy between $x/x_{MIZ} = 0.2$ and $0.4$ along the transect. While the exact location of the local maximum and maximum in wave energy between the ICESat-2 estimates and our simple model are slightly off, the difference in magnitude between them are fairly similar. Additionally, we note that the interpretation of the sharp increase in $\alpha$ around $x/x_{MIZ} = 0.5$ for the example transect shown in Fig. 4 can be readily explained by the potential misalignment between the incident wave angle and transect as well, which is discussed further in Appendix B.

The uncertainty associated with the misalignment angle and sea ice inhomogeneity implies that even when the wave direction is known, correction of wave attenuation estimates for misalignment does not necessarily provide a more reliable estimate of the wave attenuation rates, nor does it mean that negative attenuation rates should be disregarded a priori.

## 3.3 Overall attenuation statistics

The combination of sea ice inhomogeneity and measurement misalignment introduces complications in the interpretation of $\alpha$ as estimated for individual transects. However, given that there is no correlation between the ICESat-2 tracks and local sea

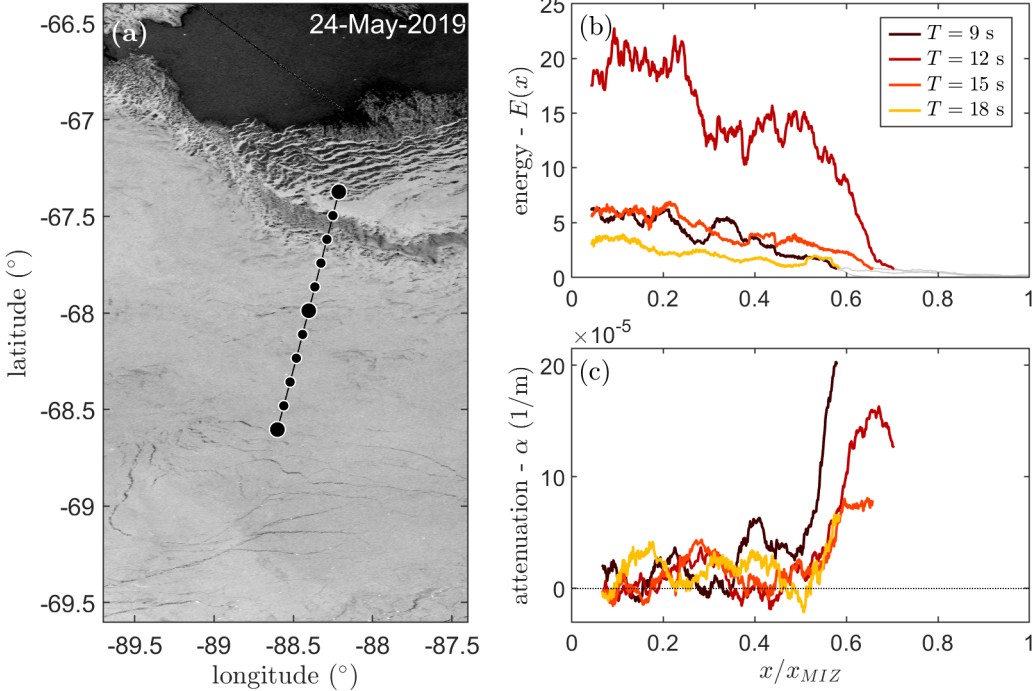

**Figure 4.** An example ICESat-2 transect (a) on 2019-05-24 with estimated spectral wave energy (b) and apparent wave attenuation rate (c). The MIZ width along the transect is given by the black line (a), which is 137.5 km wide, on top of a Sentinel-1 image from 2019-05-24 (i.e., the same day as the ICESat-2 transect). The gray lines in (b) represent observations removed after quality control.

ice and wave conditions, we may expect that for some transects the resulting fluctuations in $\alpha$ are either an underestimation or overestimation of the true wave attenuation rate. This would mean that the average apparent attenuation rate $\overline{\alpha}$ of all transects

could still provide a reasonable depiction of the mean wave attenuation rate across the MIZ. In Fig. 6a the mean apparent wave attenuation rate $\overline{\alpha}$, averaged over all transects in the dataset, is shown for $T = 12$ s as a function of $x/x_{MIZ}$. (We note that scaling with $x$ only provides no significant trend (Fig. C1a)). Here, we removed transects that contained extreme outliers in excess of three standard deviations from the median to reduce noise in the estimates of $\overline{\alpha}$ which reduced the size of the dataset by 13%. The observations suggest a linear increase in $\overline{\alpha}$ from about 0 to $10^{-4}$ across the MIZ (when plotted on logarithmic

scales, it approaches $\overline{\alpha} = 10^{-5}$ at $x = 0$, not shown). While the scatter is reasonably large, noting that the light gray area depicts the 25% and 75% percentiles, this is not necessarily surprising as some transects may present strong fluctuations, and can even become negative (e.g., see Fig. 4). Moreover, local sea ice conditions may vary significantly between transects or with respect to the 'average' sea ice conditions in the MIZ. Trends at the end and start of the MIZ are considered unreliable as few observations are available here. For $x/x_{MIZ} = 0.9$, there is a tendency for a flattening and decrease in $\overline{\alpha}$. However, we expect

this to be a methodological artifact caused by a low eSNR and sea ice morphology, i.e., near the inner limit of the MIZ the wave energy is considerably smaller such that sea ice morphology starts to contaminate the wave energy estimates.

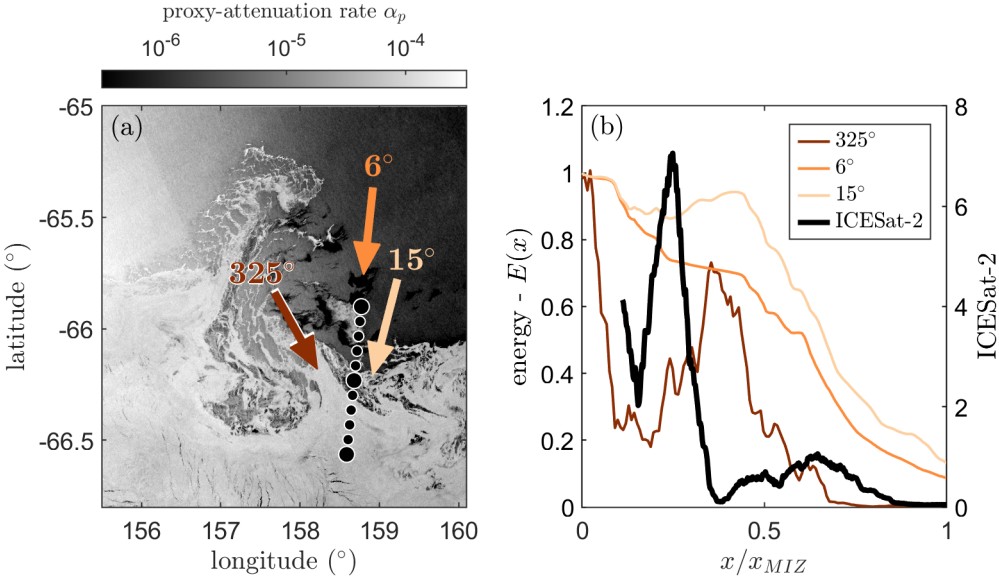

**Figure 5.** Estimated impact of misalignment angle between incident wave direction and ICESat-2 transect for example transect on 01-May-2019: (a) proxy-attenuation rate $\alpha_p$ based on Sentinel-1 backscatter (note, for illustrative purposes only); (b) estimated wave energy along the transect based on different incident wave direction. The observed wave energy for $T = 12$ s for this transect is shown in black.

In Fig. 6(b) and (c) the mean attenuation rate is compared against AMSR2-derived sea ice concentration (Spreen et al., 2008), and SMOS sea ice thickness (Huntemann et al., 2014), respectively. While we observe modest trends of $\overline{\alpha}$ with sea ice properties, they are weaker than the trend with distance into the MIZ (see Table C1). Obviously, sea ice thickness, ice concen-

tration and floe size distribution are strongly correlated with $x$ (Fig. C1b,c), and we thus suspects that these are nevertheless responsible for the increase in $\overline{\alpha}$ when scaled with the distance into the MIZ. In Fig. 6d we compare the attenuation rate against wave energy. In general, no trend can be observed with wave energy, except when $E$ is very small. This is however caused by a bias in spatial correlation with distance into the MIZ (red line), i.e., observations of low wave energy are more likely to occur deep into the MIZ, as opposed to high wave energy observations. A similar conclusion can be made when evaluating the decay

rate of the total wave energy in the MIZ (see Appendix D).

Trends of the mean wave attenuation rate for $T = 9$, 15 and 18 s are consistent with those of $T = 12$ s (Fig. 7), with all indicating that $\overline{\alpha}$ increases linearly with distance into the MIZ. The apparent attenuation rates are significantly higher for $T = 9$ s compared to the longer wave periods. While no clear sorting can be observed for $x/x_{MIZ} < 0.5$ for $T = 12$, 15 and 18 s, gradual separation in terms of frequency can be observed towards the inner MIZ. Some overlap of the 95% confidence

bounds of $\overline{\alpha}$, however, persists across the MIZ for the current dataset.

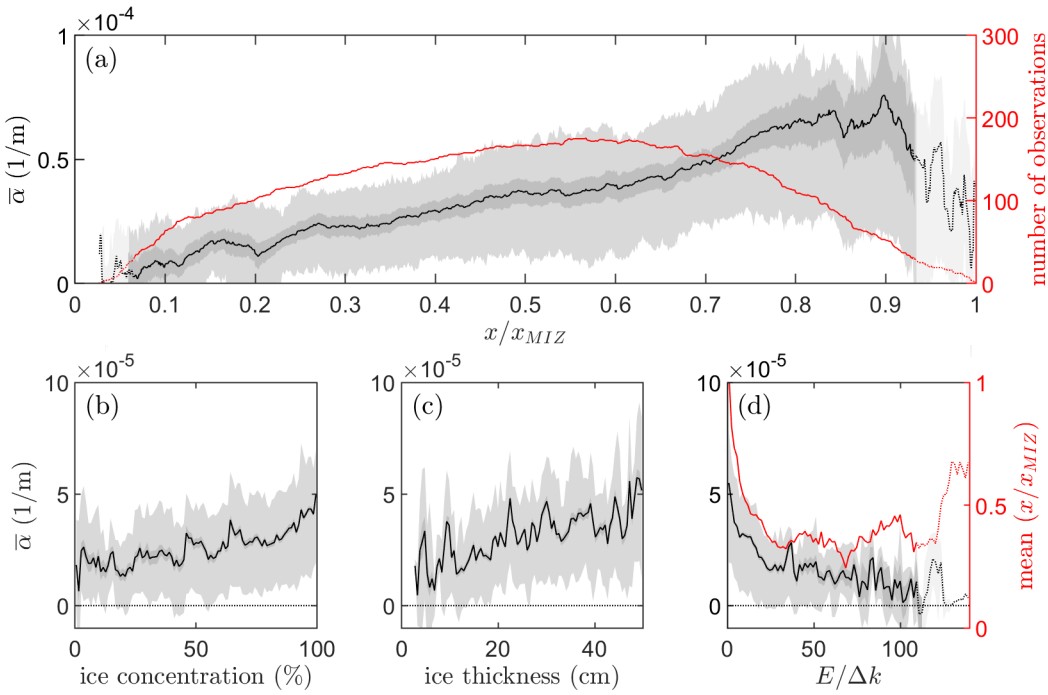

**Figure 6.** The variability of the mean apparent attenuation rate $\overline{\alpha}$ for $T = 12$ s against (a) relative distance into the MIZ, (b) sea ice concentration, (c) sea ice thickness, and (d) wave energy, using all tracks in the Fraser et al. (2024) dataset. Light gray shading corresponds to the 25th and 75th percentiles of the dataset; the dark gray shading corresponds to the 95% confidence interval of the mean (bootstrap sampling, 1000 samples with replacements). For bins with less than 30 observations, the data are shown by a dashed line, see (a) and (d).

### 3.4 Seasonality and frequency-dependence of attenuation

To assess whether $\overline{\alpha}$ varies between seasons, the mean attenuation rate for $T = 12$ s was determined for the months February, May, September and December (Fig. 8) as well. While scattered, it appears that the attenuation rate in the months of December and February increase initially much faster with distance into the MIZ, reaching a maximum already about halfway the MIZ.
Such a difference would be consistent with the expected wave climate at the ice edge of the Antarctic MIZ during the winter and summer, where wave climate is milder in summer and the sea ice is retreated from the high-energy Southern Ocean (Young et al., 2020). This is expected to lead to stronger wave-induced sea ice break-up in winter, which is expected to reduce the wave attenuation rates in the MIZ in May and September. In addition, we may expect that sea ice mechanical and material properties are different during the seasons which could potentially lead to a change in the apparent wave attenuation rate (e.g., sea ice is
thicker in summertime; Worby et al. (2008)). However, some caution is required in the interpretation of the monthly averages shown as the number of observations in February and December are very small. Thus, more data are required to confirm any seasonality of $\overline{\alpha}$.

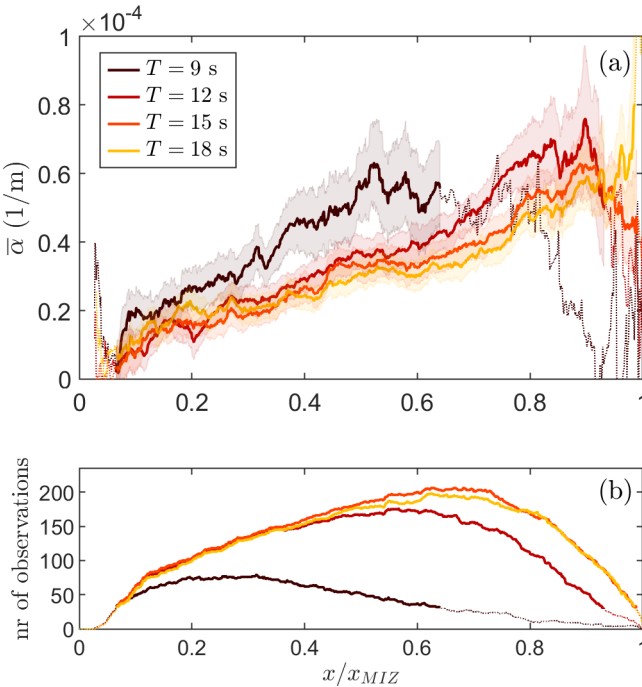

**Figure 7.** (a) The variability of the mean apparent attenuation rate $\overline{\alpha}$ for $T = 9$, 12, 15 and 18 s across the MIZ. The number of observations at each $x$ is given in (b). The shaded areas correspond to the 95% confidence interval of the mean (bootstrap sampling, 1000 samples with replacements).

To evaluate the frequency dependence of $\overline{\alpha}$, the mean apparent attenuation rate between $0.1 < x/x_{MIZ} < 0.6$ is replotted in Fig. 9 (corresponding to the region where estimates are available for all four evaluated frequencies). For $0.08 < f < 0.11$,
the attenuation rate $\overline{\alpha}$ seems to scale with a power between 2 and 3, while a considerably flatter trend may be observed for $0.08 < f$. The former is similar to the parameterization of Meylan et al. (2014), whilst the latter appears consistent with the field observations of Montiel et al. (2022) who observe significant flattening for frequencies below about 0.06 Hz. However, as the confidence limits of $\overline{\alpha}$ overlap for $T = 12$, 15 and 18 s, and the range of $f$ for which observations of $\overline{\alpha}$ are available is very limited, more data are required to ascertain such power law.

**4    Discussion & Conclusions**

The ICESat-2 observations of Fraser et al. (2024) provide a unique dataset of waves in sea ice obtained across a diverse range of Antarctic sea ice conditions (Brouwer et al., 2022). Averaging all estimates of the wave attenuation rate along the transects reveals a strong correspondence between $\overline{\alpha}$ and the relative position within the MIZ. Specifically, we find that $\overline{\alpha}$ increases linearly with distance from the ice edge. The trend with distance into the MIZ is stronger than sea ice concentration

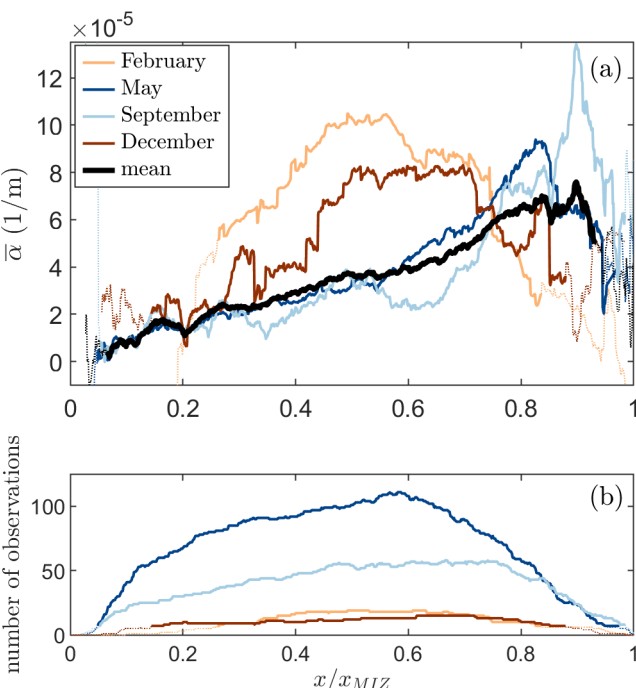

**Figure 8.** (a) The variability of the monthly mean apparent attenuation rate $\overline{\alpha}$ for $T = 12$ s across the MIZ for February, May, September and December. The number of observations at each $x$ is given in (b) noting that the observations in February and December are very limited.

or ice thickness alone (Tab. C1). The change of $\overline{\alpha}$ is expected to be a natural consequence of changing sea ice conditions with $x/x_{MIZ}$ where, on average, one may expect increases in sea ice thickness, ice concentration, and floe sizes moving into the MIZ (Fig. C1). Additionally, variability may be expected with material and mechanical properties of sea ice as well, although no observations are available on their trends with $x/x_{MIZ}$.

While the wave attenuation rates estimated using ICESat-2 measurements compare well to the rates observed by others through in-situ experiments (Meylan et al., 2014; Kohout et al., 2020; Voermans et al., 2021; Rogers et al., 2021; Montiel et al., 2022), direct validation of the ICESat-2 derived wave spectra is still required to provide greater certainty of their accuracy and the uncertainty introduced by the assumptions adopted in this study. We suspect that the largest source of uncertainty is our assumption that the dominant wave direction is in the north-south direction, i.e., well aligned with the ICESat-2 transects. While there are alternative approaches to obtain insights on the local wave direction, either from satellite observations (e.g., Hell and Horvat, 2024) or re-analysis data, the unknown uncertainty of these methods in retrieving directional information in sea ice complicates their usage. Specifically, in the study of Montiel et al. (2022), the authors find that hindcast data *"did not improve the wave growth issue and therefore motivated our choice to assume waves traveling on a north-to-south transect, for which the number of wave growth events is much more reasonable"*. This highlights the persistent uncertainty in measuring and modeling wave-ice interactions and that care should be taken in the usage of such data as a input. Thus, while the assumption that

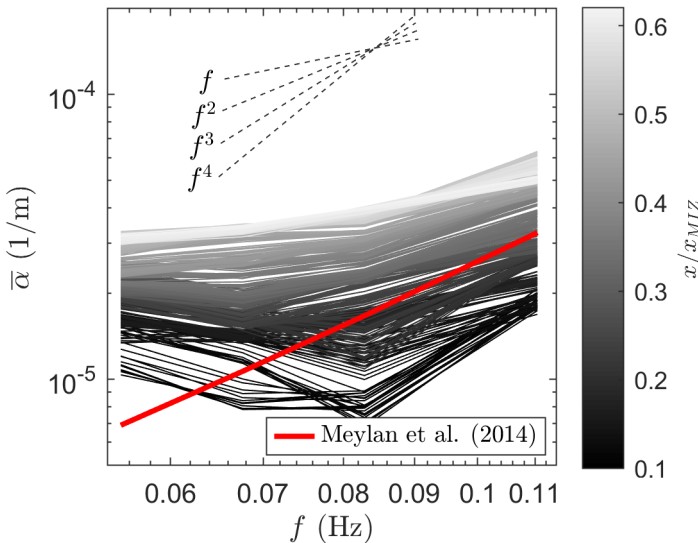

**Figure 9.** Frequency dependence of $\overline{\alpha}$ across the MIZ for $0.1 < x/x_{MIZ} < 0.62$. Comparison against the emperical model of Meylan et al. (2014) is provided.

waves are entering the MIZ from the north-south direction may therefore be acceptable and perhaps a necessary assumption, it could nevertheless lead to systematic biases in the values of the wave lengths considered, and the magnitude of the apparent attenuation rates presented in this study (see Section 2 for a rough estimate of these biases).

   Another assumption we would like to single out is the assumption of local exponential wave decay in ice covered waters that has been adopted for decades now to estimate the wave attention coefficient from experimental observations, and was used
here as well. While our observations of wave energy decay with distance from the ice edge are far from exponential (e.g., see Fig. 3), this does not disprove the validity of the assumption that wave energy decays exponential locally. Specifically, the inherent assumption in assuming exponential wave energy decay locally is that the attenuation rate is constant across the measurement section, a criterion rarely satisfied in the field. This means that as wave energy progresses further into the MIZ, the attenuation rate changes, thereby causing the wave energy decay at non-local scales to deviate from an exponential trend. Such
non-exponential behavior at non-local scales while satisfying exponential decay at local scales can be replicated by adopting a simple attenuation models. For example, using the attenuation model of Doble et al. (2015) (where $\alpha = 0.2T^{-2.13}h$) and a linear increase of sea ice thickness with $x/x_{MIZ}$, as suggested by our dataset (Fig. C1c). Naturally, while our results do not disprove the validity of local exponential decay of wave energy, it neither provides direct evidence in support of it. Direct experimental evidence for exponential decay of wave energy in sea ice at local scales is still very limited and requires significant
efforts to be validated robustly. This is by no means a simple task due to the inhomogeneity of sea ice at various length scales which would require observations with small $\Delta x$ to maintain a reasonably constant attenuation rate, but with sufficiently large $\Delta x$ to ensure the decay of wave energy is sufficiently large to be measured reliably.

The complexity of assessing wave attenuation rates from wave observations in sea ice through the reliance on traditional assumptions is further highlighted by the identified impact of the misalignment angle between the incident wave direction and the ICESat-2 transects $\Delta\theta$ and sea ice inhomogeneity on estimates of $\alpha$. Specifically, we have shown here that strong inhomogeneity of sea ice (particularly near the ice edge) and $\Delta\theta \neq 0$ can lead to significant fluctuations in the observed wave energy, which in turn leads to strong fluctuations in estimates of the wave attenuation rates. This may lead to apparent negative attenuation rates which are not necessarily a consequence of wave growth, but more likely a methodological artefact. In-situ observation methods that can achieve high spatial resolution measurements, such as sea floor cables (Smith et al., 2023), are likely to provide further insights on the accuracy of the ICESat-2 derived estimates of $\alpha$. Large-number in-situ buoy deployments in grid formation may allow assessment of the impact of $\Delta\theta$ and sea ice inhomogeneity on the estimates of $\alpha$ if $\Delta x$ is sufficiently small, however, such a deployment would be complicated by the presence of shear currents. Obviously, such data will need to be collected concurrently with the properties of sea ice across such scales to conclusively verify the accuracy of $\alpha$ in general, and the validity of the typically adopted assumptions.

While the correspondence observed between $\overline{\alpha}$ and $x/x_{MIZ}$ is empirical, it may provide alternative approaches to model wave attenuation in global models if the MIZ width is a known variable. A linear relationship of the form $\overline{\alpha} = \beta x/x_{MIZ} + \gamma$, would imply that wave energy decays into the Antarctic MIZ as:

$$E(f,x) = E(f,0)\exp(-\beta x^2...) \tag{3}$$

where $\beta$ is a wave attenuation rate coefficient which varies with wave frequency. (We note that this form can be retrieved as well from the model of Doble et al. (2015) and considering a linear function between $x$ and $h$, as supported by Fig. C1c.) The variability of the multiple sea ice properties are implicitly embedded in $x/x_{MIZ}$. Obviously, such a model ignores the presence of strong local variability of wave-ice conditions, and such an approach is therefore unlikely to model waves in an accurate way at small spatial and short time domains (Herman, 2024). Nevertheless, at global and climate scales, such detail may not necessarily be required and the model may provide a major advantage compared to current physics based models which typically rely on system variables that are not straightforward to measure, retrieve or simulate at global scales. As the scaling of $\overline{\alpha}$ depends on the MIZ width, which is defined based on the depth of wave energy propagation and typically unknown, it will require other means to parameterize $x_{MIZ}$ independently before this is feasible.

*Data availability.* The processed ICESat-2 data underlying this manuscript are freely available at https://data.aad.gov.au/metadata/AAS_4528_ICESat-2-wave-attenuation-tracks or http://dx.doi.org/doi:10.26179/q9pe-w283. AMSR2-derived sea ice concentration and SMOS derived thin sea ice thickness were obtained from the University of Bremen (https://seaice.uni-bremen.de/).

## Appendix A: Proxy-attenuation rate from Sentinel-1 backscatter

A proxy-estimate of the wave attenuation rate $\alpha_p$ was based on the Sentinel-1 backscatter $\sigma$. The distribution of the backscatter intensity from the image as presented in Fig. 5a depicts a bimodal distribution, with modes corresponding to open water (lower

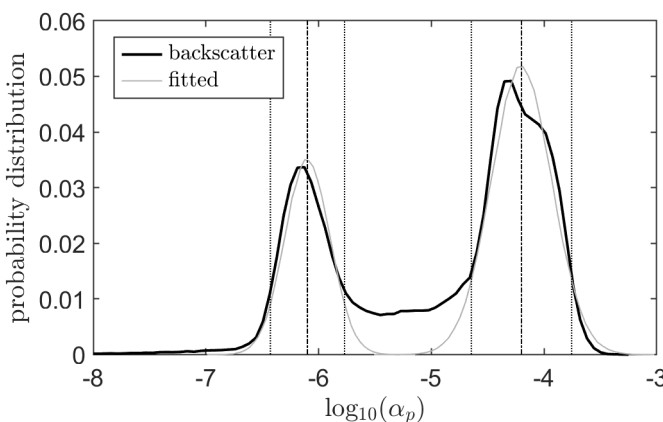

**Figure A1.** The distribution of $\alpha_p$ (black) as derived from Eq. A1 and based on the Sentinel-1 backscatter image shown in Fig. 5a. Fitted to the individual modes are normal distributions (gray), with dash-dotted lines corresponding to the mean, and dotted lines to the 5th and 95th percentiles.

mode) and sea ice (higher mode). We scaled $\sigma$ in such a way that $\alpha_p$ represents realistic values for open water and sea ice, respectively, with $\alpha_p$ given by:

$$\alpha_p = 10^{\log_{10}(\sigma^2 \times 10^{-12.5})} \tag{A1}$$

The distribution of $\alpha_p$ is shown in Fig. A1. Normal distributions were fitted to the individual modes, giving a 5th, 50th and 95th percentiles of $3.7 \times 10^{-7}$, $7.9 \times 10^{-7}$, $1.7 \times 10^{-6}$ for open water, and $2.3 \times 10^{-5}$, $6.3 \times 10^{-5}$, $1.7 \times 10^{-4}$ for sea ice. These values correspond well to the expected range of $\alpha$ as derived from in-situ buoy observations (e.g., Meylan et al., 2014; Voermans et al., 2021; Montiel et al., 2022), and open water attenuation rates of swell (Jiang et al., 2016; Babanin, 2012). We reiterate, however, that Eq. A1 cannot be used in practical modeling applications and is used in this study solely to illustrate the impact of wave and measurement misalignment. This is evident by the gradient observed in $\sigma$ at 65°S in Fig. 5a yet may all be interpreted as open water.

## Appendix B: Misalignment angle, example transect Fig. 4

In the example transect shown in Fig. 4 a steep increase in the estimated wave attenuation rate $\alpha$ at around $x/x_{MIZ} = 0.5$ was observed. Notably, this steep increase cannot be linked straightforwardly to an obvious change in sea ice conditions based on visual inspection of the corresponding Sentinel-1 image. We consider here the misalignment angle of the incident wave direction and the transect and the inhomogeneity of the sea ice as a likely reason of the steep increase in $\alpha$. In Fig. B1 a larger spatial area of the sea ice conditions is shown, revealing strong variability in the ice edge position. Particularly, the region of sea ice surrounding 67°S 90°E is expected to lead to a sharp drop in wave energy around $x/x_{MIZ} \approx 0.5$ if the incident wave direction is approximately $345°$ (relative to north). That is, waves arriving along the transect at $x/x_{MIZ} < 0.5$ will have crossed

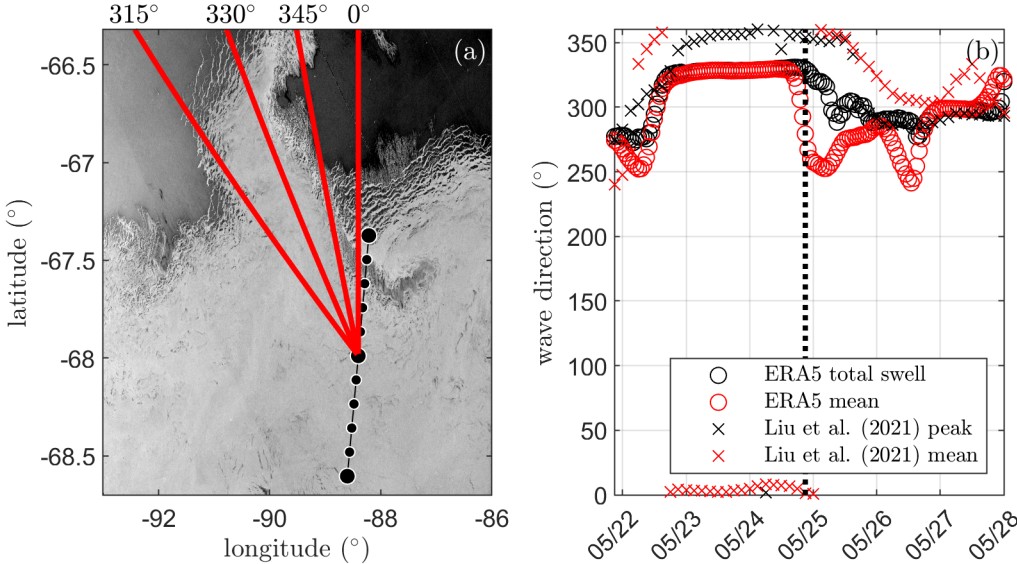

**Figure B1.** Impact of sea ice inhomogeneity and misalignment between incident wave direction and ICESat-2 transect (an expansion of the region shown in Fig. 4). (a) Sentinel-1 image from 2019-05-24, (b) wave direction from hindcasts.

a significantly shorter distance through sea ice in comparison to the wave energy arriving at $x/x_{MIZ} > 0.5$. Wave directional data just north of the observation site (65.75°S 89°E) obtained from ERA5 Reanalysis and WAVEWATCHIII hindcast Liu et al. (2021) suggests an incident wave direction of around $345°$ is well within the range of possibility.

**Appendix C: Comparing sea ice and wave attenuation statistics**

330

In Fig. C1a the mean apparent attenuation rate $\overline{\alpha}$ is compared against the distance from the ice edge $x$. We note that scaling $\overline{\alpha}$ with $x$ leads to a considerable flattening of the trend in comparison to the scaling with $x/x_{MIZ}$ (i.e., Fig. 6a), which is reflected by the coefficients of determination of $r^2 = 0.25$ and $r^2 = 0.96$, respectively.

335

In Tab. C1 the coefficients of determination between various variables are shown. Most notably, the statistics imply that $\overline{\alpha}$ shows stronger correlation with $x/x_{MIZ}$ than either $\overline{C}$ or $\overline{h}$. Both $\overline{C}$ and $\overline{h}$ are strongly correlated with $x/x_{MIZ}$ (Fig. C1 and Tab. C1) which seems to imply that it is not an individual sea ice property that defines the wave attenuation rate, but a combination thereof. We note that interpretation of Fig. C1 and the corresponding $r^2$ value requires some caution, as the SMOS derived sea ice thickness has a maximum of 50 cm, which means that taking the average is likely to cause a bias in the mean sea ice thickness trend as indicated by the 75th percentile.

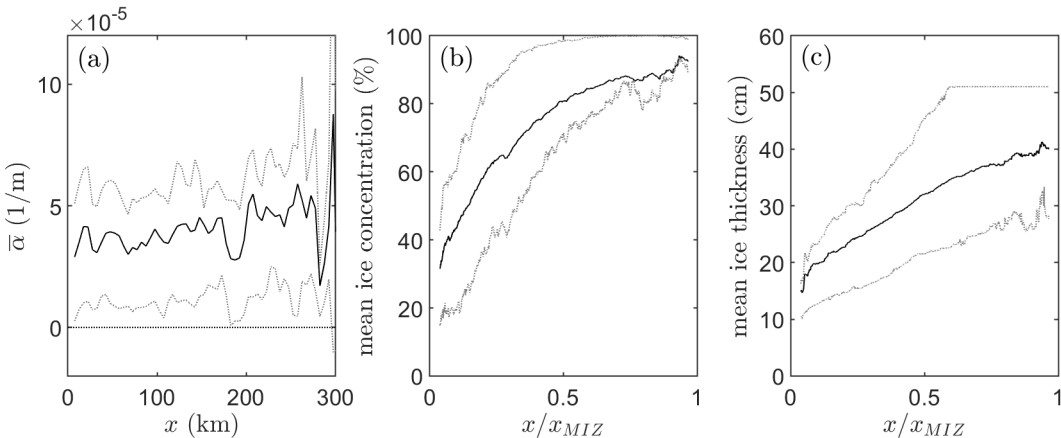

**Figure C1.** (a) The variability of the mean apparent attenuation rate $\overline{\alpha}$ for $T = 12$ s against the distance from the ice edge $x$, and the mean ice concentration (b) and sea ice thcikness (c) against the normalized distance into the MIZ. Dotted lines correspond to the 25th and 75th percentiles of the dataset. We note that the sea ice thickness observations are restricted to a maximum thickness of 50 cm, and care should be taken in the interpretation of the mean.

|  | $\alpha$ | $\overline{\alpha}$ | $\overline{C}$ | $\overline{h}$ |
|---|---|---|---|---|
| $x/x_{MIZ}$ | 0.12 | 0.96 | 0.88 | 0.98 |
| $C$ | 0.06 | 0.58 | - | - |
| $h$ | 0.06 | 0.68 | - | - |

**Table C1.** Coefficients of determination $r^2$ between system variables, with $\alpha$, $h$, $C$ and $x/x_{MIZ}$ being the apparent attention coefficient, ice thickness, ice concentration and relative distance within the MIZ. Overbar indicates averaging.

## Appendix D: Attenuation rate of significant wave height

Although the wave attenuation rate is frequency dependent, the rapid attenuation of wave energy at higher frequencies means that the spectrum becomes narrower deeper into the MIZ. In such a case, one may look at the attenuation rate of the significant wave height $\alpha_H$ instead, which has the advantage of being more robustly measured than the wave spectrum but at the cost of the error made by ignoring the frequency dependence (Kohout et al., 2020):

$$\frac{dH}{dx} = -\frac{\alpha_H}{2} H_s \tag{D1}$$

where $H_s = 4\sqrt{m_0}$ is the significant wave height and $m_0$ is the first order moment of the wave spectrum. While the dataset suggests that the wave attenuation rate of $H_s$ decays strongly with $H_s$ (Fig. D1a), this is in large part biased by the strong correlation between $H_s$ and the relative position within the MIZ (Fig. D1c,d) (see also Kohout et al., 2020). That is, low $H_s$ is predominantly observed deep into the MIZ where sea ice concentration and sea ice thickness are relatively high, whereas larger $H_s$ tends to be observed near the ice edge where sea ice concentration and sea ice thickness tends to be low.

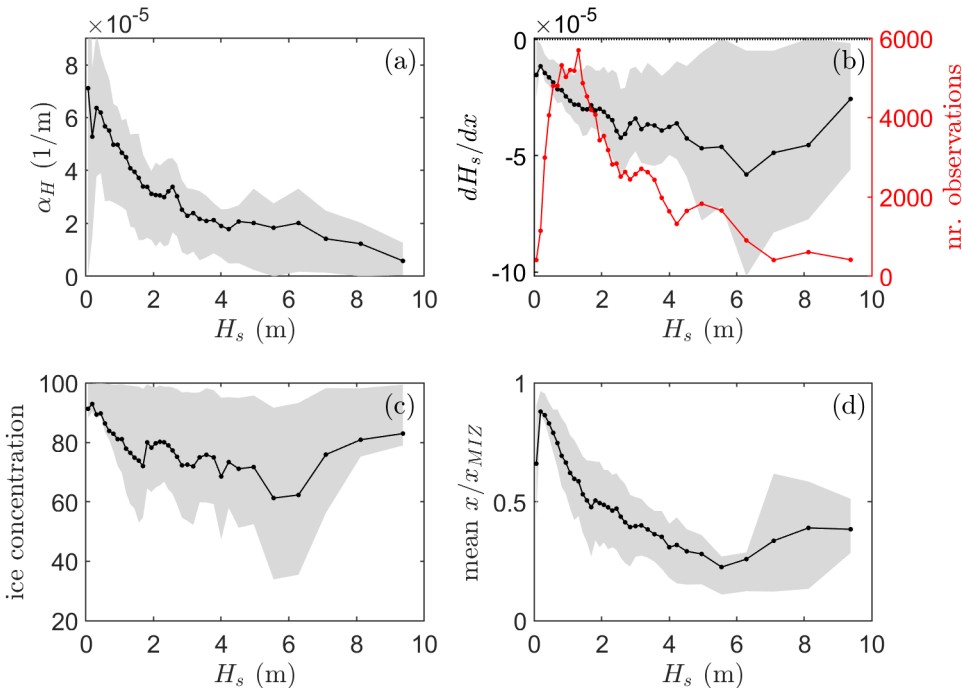

**Figure D1.** Attenuation rate of significant wave height (a), wave height gradient (b), mean sea ice concentration (c) and relative position in the MIZ (d) against significant wave height for all transects.

*Author contributions.* Conceptualization: ADF. Methodology: JJV. Formal analysis: JJV, ADF. Data interpretation: all authors. Writing - Original Draft: JJV. Writing - Review & Editing: all authors.

*Competing interests.* The contact author has declared that neither they nor their co-authors have any competing interests

*Acknowledgements.* This project is supported through the Australian Government as part of the Antarctic Science Collaboration Initiative program and contributes to Project 6 (Sea Ice) of the Australian Antarctic Program Partnership (project ID ASCI000002). ADF is supported by the Australian Research Council through a Future Fellowship (FT230100234). ADF acknowledges the generous support of the Harris Charitable Trust through the Antarctic Science Foundation. QL acknowlegdes the National Natural Science Foundation of China (42106012), the Shandong Provincial Natural Science Fund for Excellent Young Scientists Fund Program (Overseas) (2023HWYQ-056), the Taishan Scholars Program (tsqnz20221111) and the Fundamental Research Funds for the Central Universities (202441007). AVB acknowledges support of the Centre for Disaster Management and Public Safety (CDMPS). Authors acknowledge the usage of EO Browser, Sinergise Solutions d.o.o., a Planet Labs company, in retrieving Sentinel-1 imagery. Authors would like to thank Tripp Collins for feedback on this study.

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
