# Peer review of "Finely-resolved along-track wave attenuation estimates in the Antarctic marginal ice zone from ICESat-2"

_EGUsphere, 2024_

## Referee Comment (RC1)

In their manuscript „Finely-resolved along-track wave attenuation…", Joey Voermans and coauthors perform an analysis of the apparent wave attenuation in the Antarctic MIZ based on satellite (ICESat-2) derived along-track profiles of wave energy $E(x)$ for four selected wave frequency bands (corresponding to wave periods of 9, 12, 15 and 18 s). From the original data, published earlier in Brouwer et al., 2022, the Authors extract 320 high-quality $E(x)$ profiles and, assuming exponential attenuation, for each of them compute the corresponding profile of the attenuation coefficient $\alpha(x)$. Based on that data, several statistics of $\alpha(x)$ are computed, including the average $\bar{\alpha}(x)$ for all profiles and for individual months, as well as relationships between $\alpha(x)$ and ice thickness, concentration and $x/x_{\text{MIZ}}$ (i.e., the distance from the ice edge scaled with MIZ width). An important part of the analysis is related to the influence of spatial variability of sea ice and wave conditions, and of non-zero angles between the satellite tracks and the wave propagation directions on the observed shapes of $E(x)$.

1. My major comment on the manuscript is this:
   I fully agree with the Authors that they have at their disposal "a unique dataset of waves in sea ice obtained across a diverse range of Antarctic sea ice conditions" (lines 222-223). Going from a few (in many cases: only two) data points to high-resolution profiles of wave energy is a big step forward. The new dataset makes it possible to find new patterns in the data, but also to verify assumptions that have been commonly used so far in analyses of wave attenuation in sea ice. However, to fully use the potential offered by the new data, new approaches are necessary – whereas my overall impression when reading the manuscript was that the Authors tried their best to make their study methodologically as close to the previous ones as possible, and to keep all old assumptions untouched.
   Neither the individual $E(x)$ profiles nor the final, average attenuation $\bar{\alpha}$ are exponential. In particular, none of the $E(x)$ curves presented in the figures resembles an exponential curve; in some cases, like e.g. in Fig.1a, the $E(x)$ profile is concave rather than convex, so that the exponential approximation is particularly inadequate. Still, the Authors decide to "use of the commonly adopted assumption that wave energy decays exponentially" (line 105): their analysis begins with piecewise approximation of $E(x)$ with exponential functions. First, as just said, even a quick look at the data is enough to say that this is a poor choice, and second, given the very high spatial resolution of the profiles, no a priori assumptions regarding their shapes are necessary. To the contrary, the shapes can be found as a result of the analysis. Several interesting questions could be answered this way. For instance, is there a one type of function that provides a satisfactory fit to the majority (or a large subset) of the analyzed profiles? Is the fit generally better in the inner MIZ than close to the ice edge? Which function provides a good fit to the average energy profile? Is it really exp(-bx$^2$), as the analysis based on exponential attenuation suggests? If yes, how does this function perform for individual profiles?
   In previous studies, assumptions regarding the shape of E(x) were necessary given the large spacing between data points, and the choice of exponential shape seemed natural given what we know from theory about attenuation related to individual physical processes. However, the Authors themselves convincingly show in Section 3.2 and Fig. 4 (and 3 as well) that due to many different factors the apparent attenuation observed along "random" satellite tracks, and attenuation that would be observed along wave trains undergoing a certain physical process, are two very different things.
   In short, in my opinion the analysis can be made more convincing and more valuable for future applications if the Authors rethink their approach to the data and modify the manuscript accordingly.

2. I have doubts regarding the widths of the MIZ used in the analysis. The Authors say they used $x_{MIZ}$ estimates from Brouwer et al (2022), which are "based on the depth of wave penetration into the MIZ" (line 85), but Fig. 2 and, especially, 3 suggest that these estimates might be inadequate for the present purpose. In Fig. 3b, a reasonable estimate of $x_{MIZ}$ seems to be between 0.6 and 0.7 of the actually used one – which makes a huge difference. At first, I thought that maybe this is just an unfortunately selected example, but the drop in the amount of valid data points in the inner MIZ (red line in Fig.5a) suggests that $x_{MIZ}$ used in the analysis is systematically overestimated. Wouldn't it be more reasonable and consistent to compute $x_{MIZ}(f)$ as the largest $x$ for which $E$ can be reliably estimated (i.e., for non-grey parts of Fig. 2b and 3b), and only then take a median of those values?
The proper estimation of $x_{MIZ}$ is very important, because the whole analysis is performed in terms of $x/x_{MIZ}$. In its present form, considering what is presented in Figs. 2, 3 and 5, it is not very convincing.

3. One of the main results of this analysis is the average (over all profiles) attenuation in function of $x/x_{MIZ}$: it is found that, if exponential attenuation is assumed, it increases linearly with $x/x_{MIZ}$. This result is interesting, but it is hard to think how it could be used in practice (as the Authors suggest in the discussion). First, the relationship $\alpha(x/x_{MIZ})$ is useful only if $x_{MIZ}$ is known – which is possible only if we already know the waves! Notably, replacing the wave-based definition of the MIZ with another one (e.g., ice concentration based one) would make the whole approach inconsistent, so it doesn't provide a solution to this problem (there are studies that show that MIZ width estimates based on different criteria can be very different). Second, this approach seems reasonably simple only in 1D. In 2D, given that the outer and inner boundaries of the MIZ are irregular curves evolving in time, using the scaled coordinate perpendicular to those boundaries would require solving Laplace's equation (as in Strong et al., JPO, 2017) at each model time step. An approach that is anything but simple when one thinks about the details.
I think therefore that the statement "it may provide alternative approaches to model wave attenuation in global models if the MIZ width is a known variable" is very misleading.

4. I have a question regarding the data in Fig. 5a,b,c. In panels b and c, there are hardly any values of $\alpha$ larger than, say, $5\ 10^{-5}\ m^{-1}$. In panel a, values above $5\ 10^{-5}\ m^{-1}$ are common in the inner MIZ ($x>0.7x_{MIZ}$). Are there no ice thickness and concentration data from those regions, so that they don't appear in b and c? Or what is the reason for the different ranges of $\alpha$ in different plots?
The increase of attenuation with ice concentration would be much steeper if all data were included, wouldn't it?

5. In spectral wave models, $c_g dE/dx \sim AS_{ice} + S_{ee}$ ($c_g$ - group velocity, $A$ – ice concentration; 'ee' – everything else). If we forget 'everything else' for a moment, a measure of $S_{ice}$ can be obtained by dividing attenuation with ice concentration. I wonder how those profiles would look like. How much of the observed increase of $dE/dx$ with distance from the ice edge can be attributed to the increase in $A$? Is there a zone of high $S_{ice}$ close to the ice edge, as suggested by some studies?
(A note related to point 1 above: even if we expect that $S_{ice}$ represents a single process leading to exponential attenuation, $dE/dx$ should be "corrected" for ice concentration before it can be treated as an estimate of $S_{ice}$.)

---

## Referee Comment (RC2)

**Review egusphere-2024-2104**

The paper estimates wave energy attenuation rates from preprocessed IS2 tracks around Antarctica. It discusses the advantages and caveats of such an estimate and provides the wave attenuation coefficient as a function of non-dimensional MIZ distance. It paper is well-written, has a clear outline, and derives plausible attenuation coefficients of wave energy in the MIZ around Antarctica. It provides evidence that wave attenuation is a function of frequency. However, a few major points must be raised after reviewing the manuscript:

1. The paper uses a preprocessed data set from Brouwer et al. 2022 and Fraser et al. 2024 that is based on the ATL07 product. One ATL07 segment contains the median (or mean) of 150 photon retrievals, which leads to varying segment lengths depending on the photon density of the data. The paper re-samples the segment heights of varying lengths to 8m, which can lead to substantial aliasing of the wave signal, especially in marginal ice zones where the photon retrieval can be below, but wave amplitudes are high. That is not discussed.
   In addition, the most recent versions of ATL07 filter the wave signal on purpose to remove surface wave effects, as they are considered noise for sea ice products. The use of other, lower-level products would be necessary when this analysis is applied at scale. (see Issue 3 in version 6 if the data release: https://nsidc.org/sites/default/files/documents/technical-reference/icesat2_atl07_atl10_known_issues_v006.pdf)

2. Sample uncertainty
   In one same paragraph (L89ff), they describe the section length L as 128 data points and/or a section length of 2048, 8192, or 16384 meters, but an L with 128 data points with 8-meter spacing results in 1024 meters. This needs to be clarified.
   Further, they say the choice of L is arbitrary but then acknowledge that the scatter between estimates is "reasonably large" (L179). A better quantification and accounting for how many samples one would need per wavenumber for a good estimate would strengthen the paper. The impact of ice edges and step-like changes in sea ice height on the FFT is also entirely ignored, which can substantially impact the estimated wave spectra when doing FFT (Hell and Horvat, 2024).

3. angle projection uncertainty
   The authors establish the projection of the true wave number $k_a$ on the observed wave number $k$ through the incident angle $\theta$ but then state that they follow the "common assumption" of waves propagating in the direction of the IS2 tracks. The authors argue that the circumpolar nature of Antarctica favors waves in the north-south direction. I'm afraid I have to disagree with the argument, and I would like to see evidence for that statement. Think it is the most common that the incident angle must be addressed when measuring spectra from IS2 because the main wind/storm direction is east-west. The

median zonal wind direction over the southern ocean is about 5 m/s, while the mean meridional wind direction is near zero. In other words, it would be surprising if the dominant incident wave energy comes from the north in line with IS2 tracks. We would expect a mean wave direction going southeastward, leading to systematic biases in the wavenumber estimates. The wave climate may also substantially vary by region as the southern ocean storm has a clear climatological pattern (Hoskins and Hodge, 2005, for example). The assumption that wave travels in the north-south direction might be common, but there is evidence that using this as a general assumption for analyzing IS2 around AA is wrong.

Section. 3.2:
The study then tries to access the impact of wave direction on the attenuation rate using some test cases using Sentinel-1 SIC as a tracer for the \alpha. What is \alpha_p? This variable is not introduced. Further, while it is plausible to model variation in $\alpha$ as a function SIC, the functional relation between $\alpha$ and SIC is not given.

The authors realize that even slide incident angles will create different wave amplitudes along the transect due to the strong sea ice heterogeneity along the propagation path. While I share their statement about the resulting uncertainty of the wave attenuation rates, I don't follow their argument about negative attenuation rates, i.e., wave energy growth. Without energy input, wave action can only stay constant or decay with distance into the sea ice; however, here, the quantity used is wave energy, which can increase while the action is conserved. This can be done by wave-current or wave-sea ice interaction (Squire, 2018, or similar). The discussion about "negative attenuation rates" is not very physical without adhering to these wave actions and is confusing to the reader.

3. Sample tracks
   section 3.3: Here, the authors try to give a best guess of the overall attenuation rate from all transects derived in Brouwer et al. The Authors hope that the noise (or randomness in direction) cuts down enough that the mean is a reliable estimate of the attenuation rates. They do not discuss or quantify: a) how many transects they use, b) how those are spatially distributed, or c) what criteria are chosen to select these tracks. It is then questionable if this dataset is a representative sample of wave attenuations around AA, even though the text suggests that. This section needs more context on how representative this sample is. The reader would need to use two other papers to get that information —Brouwer et al derived 304 tracks in 4 months of one year. Given the amount of data IS2 provides, this is then a small test set of data samples with substantial uncertainties in the underlying metrics.

4. Unknown error due to unknown other metrics
   The attenuation rates are estimates in the normalized distance x/x_miz, while how x_miz is derived is not described. (likely defined in one of the other papers). This metric is

important because of the misalignment and sampling uncertainties, the attenuation rates will also depend on the robustness of the x_miz measure, as this appears in the denominator of \alpha. Uncertainties in x_miz can have a large impact on the estimated attenuation rate.

From section 3 in Brouwer et al. the x_miz is based on co-aligned daily SIC products of 6.25km depending on the total length of x_miz (not given, but often less than 50 km). Could this lead to additional substantial biases in the attenuation estimate? The authors should be more explicit about what is done here and what the impacts are.

Despite the paper's shortcomings, the paper provide evidence that attenuation rates vary with frequency and with distance in the MIZ. These are new finding for IS2 observation but not in general (Meylan et. al 2014, Thomson et al 2021, MONTIEL et al. 2022, and a few others). In summary, even though it is novel to derive attenuation rates from IS2, this paper has methodological flaws that leave questions about the accuracy and validity of the estimate, as discussed in other publications already (Hell and Horvat, 2024).

If this paper wants to describe a new method for calculating attenuation rates, the stated concern give reason why this method is problematic; if the paper wants to derive actual attenuation rates for later use, it needs to qualify its sampling and give reason why this estimate is robust.

additional remarks

- I would reword the statement in L.6 that this samples "over a wide range of sea ice condition".
- L 161: "completely different attenuation rates" - that is strong wording, I would remove that.
- L 244: How do we see wave-current interaction in figure 4? I don't follow
- Fig.8: the coloring choice is unfortunate.

References:

Hoskins, B. J., and K. I. Hodges, 2005: A New Perspective on Southern Hemisphere Storm Tracks. *J. Climate*, **18**, 4108–4129, https://doi.org/10.1175/JCLI3570.1.

Squire, V. A., 2018: A fresh look at how ocean waves and sea ice interact. *Phil. Trans. R. Soc. A.*, **376**, 20170342, https://doi.org/10.1098/rsta.2017.0342.

---

## Author Comment (AC1)

Reviewer 2

The paper estimates wave energy attenuation rates from preprocessed IS2 tracks around Antarctica. It discusses the advantages and caveats of such an estimate and provides the wave attenuation coefficient as a function of non-dimensional MIZ distance. It paper is well-written, has a clear outline, and derives plausible attenuation coefficients of wave energy in the MIZ around Antarctica. It provides evidence that wave attenuation is a function of frequency. However, a few major points must be raised after reviewing the manuscript:

1. The paper uses a preprocessed data set from Brouwer et al. 2022 and Fraser et al. 2024 that is based on the ATL07 product. One ATL07 segment contains the median (or mean) of 150 photon retrievals, which leads to varying segment lengths depending on the photon density of the data. The paper re-samples the segment heights of varying lengths to 8m, which can lead to substantial aliasing of the wave signal, especially in marginal ice zones where the photon retrieval can be below, but wave amplitudes are high. That is not discussed.
In addition, the most recent versions of ATL07 filter the wave signal on purpose to remove surface wave effects, as they are considered noise for sea ice products. The use of other, lower-level products would be necessary when this analysis is applied at scale. (see Issue 3 in version 6 if the data release: https://nsidc.org/sites/default/files/documents/technicalreference/icesat2_atl07_atl10_known_issu es_v006.pdf)

Based on the reviewer's comment, we have sub-sampled the 8 m resolution signal to a resolution of 16 and 24 m to identify the impact of the sample length on the average attenuation rate $\bar{\alpha}$. We refer to the figure below for the comparison for different wavelengths. As the observed differences are very small, we don't think that the issue highlighted has implications for our results nor the interpretation thereof.

We are also aware of the changes to the ATL07 filter now unfortunately excluding the MIZ. We suggest that SlideRule (https://slideruleearth.io/) could be used to produce heights similar to the earlier ATL07 releases, although we have not yet investigated this.

[Figure]

2. Sample uncertainty

In one same paragraph (L89), they describe the section length L as 128 data points and/or a section length of 2048, 8192, or 16384 meters, but an L with 128 data points with 8-meter spacing results in 1024 meters. This needs to be clarified.

We have used 128 sample points (1024 m) as the window, but used longer section lengths L to obtain smoother statistics. We acknowledge that this may not be clear in the manuscript and will rephrase the corresponding sentences.

Further, they say the choice of L is arbitrary but then acknowledge that the scatter between estimates is "reasonably large" (L179). A better quantification and accounting for how many samples one would need per wavenumber for a good estimate would strengthen the paper.

Indeed, the choice of L is arbitrary, as per the reasons discussed in section 2. The scatter referred to in L179 is not necessarily related to the arbitrariness of L, but more likely to the variability of sea ice conditions between ICESat-2 transects.

For the impact of $L$ on the results we refer to Fig. 1c in the manuscript, specifically, comparing the red and blue lines. Steep changes in wave energy around x=90 km causes loss in spatial detail due to increases in number of samples per wave number. This is, however, not everywhere the case, and an optimum number of samples is therefore unlikely to be a fixed number. Without ground truth observations of the wave spectra or wave attenuation, it is unlikely that a scientific based optimal number can be provided for the quantification of wave energy.

The impact of ice edges and step-like changes in sea ice height on the FFT is also entirely ignored, which can substantially impact the estimated wave spectra when doing FFT (Hell and Horvat, 2024).

In this study we adopted the definition of the MIZ as proposed by Brouwer et al., (2022). In physical terms, this definition may be interpreted as the position where the surface elevation variance transitions from wave dominated to ice dominated. We expect that if the sea ice freeboard (or any other feature of sea ice) introduces substantial errors in the estimates of the wave spectra, this will be outside the MIZ as defined by Brouwer et al.

We did a simplified test to see what happens to the estimated wave energy with distance into the MIZ by taking a simple monochromatic wave and adding artificial sea ice contours on top. We approximated the sea ice by means of 500 ice floes, where the length of the floes increases and the open water between floes decreases with distance into the MIZ. The ice thickness was assumed to increase linearly with distance into the MIZ from 10 cm to 5 m, and the freeboard was assumed to be 10% of the ice thickness.

In the two figures below, we show two different sea ice cases. For the first, the distance between ice floes varies between 300 and 30 m, the second it varies from 100 to 0.1 m. We can see that only for the first, and for very low incoming wave energy, the sea ice has impact on spectrum estimates. In the top left subplot, we may see a strong deviation from the expected curve (wave dominated to sea ice dominated variance). In the other cases, we don't see clear deviations from the wave only case, and as such, we don't expect differences in wave attenuations estimates. For the only case that is impacted, such a transition would have been marked as the end of the MIZ. While this analysis is based on a major simplification of sea ice, we don't expect that sea ice freeboard has significant impact on our results given the currently used definition of the MIZ.

[Figure]

[Figure]

3. angle projection uncertainty

The authors establish the projection of the true wave number k_a on the observed wave number k through the incident angle \theta but then state that they follow the "common assumption" of waves propagating in the direction of the IS2 tracks. The authors argue that the circumpolar nature of Antarctica favors waves in the north-south direction. I'm afraid I have to disagree with the argument, and I would like to see evidence for that statement. Think it is the most common that the incident angle must be addressed when measuring spectra from IS2 because the main wind/storm direction is east-west. The median zonal wind direction over the southern ocean is about 5 m/s, while the mean meridional wind direction is near zero. In other words, it would be surprising if the dominant incident wave energy comes from the north in line with IS2 tracks. We would expect a mean wave direction going southeastward, leading to systematic biases in the wavenumber estimates. The wave climate may also substantially vary by region as the southern ocean storm has a clear climatological pattern (Hoskins and Hodge, 2005, for example). The assumption that wave travels in the north-south direction might be common, but there is evidence that using this as a general assumption for analyzing IS2 around AA is wrong.

We agree with the reviewer that wave energy travels along the north-south direction may not necessarily be true. Unfortunately, it is not straightforward to obtain such information. As discussed in the same section, we acknowledge that wave hindcast data may provide guidance but comes with uncertainties by itself, where different datasets can provide distinctly different values. Ultimately, only in-situ observations can provide ground truth estimates of the associated uncertainties and biases, as discussed in Section 4.

To acknowledge the potential bias from the missing wave incident angle, we will edit the corresponding paragraph to improve clarity on this matter.

Section. 3.2:

The study then tries to access the impact of wave direction on the attenuation rate using some test cases using Sentinel-1 SIC as a tracer for the \alpha. What is \alpha_p? This variable is not introduced. Further, while it is plausible to model variation in $\alpha$ as a function SIC, the functional relation between $\alpha$ and SIC is not given. The authors realize that even slide incident angles will create different wave amplitudes along the transect due to the strong sea ice heterogeneity along the propagation path. While I share their statement about the resulting uncertainty of the wave attenuation rates, I don't follow their argument about negative attenuation rates, i.e., wave energy growth. Without energy input, wave action can only stay constant or decay with distance into the sea ice; however, here, the quantity used is wave energy, which can increase while the action is conserved. This can be done by wave-current or wave-sea ice interaction (Squire, 2018, or similar). The discussion about "negative attenuation rates" is not very physical without adhering to these wave actions and is confusing to the reader.

In Section 3.2 we aim to show that the misalignment between the incident wave angle and ICESat-2 transect, and sea ice heterogeneity can lead to apparent negative attenuation rates. These are not physically real (i.e. wave growth), but methodological artefacts. Waves measured at different positions along the transect may have experienced critically different sea ice conditions, which may appear as negative attenuation rates when processing the data, but they are not (necessarily) physical.

In L150 we define $\alpha_p$ as the proxy attenuation rate. The functional relation of $\alpha_p$ is not parametrically provided but its values can be distilled from the colour bar above Figure 4a. The reason why it was not provided is because SAR backscatter images are a poor estimator of the true local attenuation rate and to avoid general adoption in future studies. Particularly, the choice of $\alpha_p$ is merely to demonstrate the methodological artefact of negative attenuation rates that may appear in the attenuation curves derived for individual ICESat-2 transects.

However, to improve readability of this section we will provide the functional relation in the main text and the considerations of the relation in an appendix. Further, to avoid confusion in the interpretation of this section, we will rephrase parts of section 3.2.

3. Sample tracks

section 3.3: Here, the authors try to give a best guess of the overall attenuation rate from all transects derived in Brouwer et al. The Authors hope that the noise (or randomness in direction) cuts down enough that the mean is a reliable estimate of the attenuation rates. They do not discuss or quantify: a) how many transects they use, b) how those are spatially distributed, or c) what criteria are chosen to select these tracks. It is then questionable if this dataset is a representative sample of wave attenuations around AA, even though the text suggests that. This section needs more context on how representative this sample is. The reader would need to use two other papers to get that information —Brouwer et al derived 304 tracks in 4 months of one year. Given the amount of data IS2 provides, this is then a small test set of data samples with substantial uncertainties in the underlying metrics.

We acknowledge the reviewer's sentiment that more data is better. Nevertheless, we believe that 304 tracks provides a large enough dataset to derive trends in wave attenuation (i.e., see uncertainty

bounds in Figures 6 and 7). There is, however, a seasonal bias in the observations as we have considerably more transects available from May and September (see section 3.4).

We note that the number of observations are provided throughout the manuscript, and like to refer the reviewer to Fig. 5a, where in red (see right axis) the number of observations are provided. In Figures 6 and 7 they are provided in the subplots (b).

To provide an overview of the spatial distribution of the observations, we will include a map of the Antarctic continent and the distribution of the corresponding transects as a figure in the manuscript.

4. Unknown error due to unknown other metrics The attenuation rates are estimates in the normalized distance x/x_miz, while how x_miz is derived is not described. (likely defined in one of the other papers). This metric is important because of the misalignment and sampling uncertainties, the attenuation rates will also depend on the robustness of the x_miz measure, as this appears in the denominator of \alpha. Uncertainties in x_miz can have a large impact on the estimated attenuation rate. From section 3 in Brouwer et al. the x_miz is based on co-aligned daily SIC products of 6.25km depending on the total length of x_miz (not given, but often less than 50 km). Could this lead to additional substantial biases in the attenuation estimate? The authors should be more explicit about what is done here and what the impacts are.

The MIZ is defined by means of the depth of wave penetration into the sea ice and has a significantly finer resolution than that of SIC products. While this is briefly mentioned in Section 2, we acknowledge that the manuscript can improve with further clarification on this definition.

Additionally, the majority of the transects considered here are from May and September, where the MIZ is about 200 km wide. An error of 6.25 km in the ice edge position is unlikely to cause considerable bias.

Despite the paper's shortcomings, the paper provide evidence that attenuation rates vary with frequency and with distance in the MIZ. These are new finding for IS2 observation but not in general (Meylan et. al 2014, Thomson et al 2021, MONTIEL et al. 2022, and a few others).

The authors respectfully disagree with the reviewer's view that the findings are not new in general. Neither Meylan et al. 2014 nor Thomson et al. 2021 identifies nor discusses explicitly wave attenuation rates as a function of distance into the MIZ, or any parameters that may be considered to be strongly correlated to $x/x_{MIZ}$. While Montiel et al. 2022 discusses changing attenuation rates in terms of sea ice concentration, this is not necessarily reflective of $x/x_{MIZ}$. Additionally, in Montiel et al., no straightforward variability of $\alpha$ and sea ice concentration is found.

Whereas it is well known that wave attenuation tends to increase with sea ice thickness, sea ice concentration, floe size etc, and each of these are likely correlated with the distance from the ice edge or latitude, to the best of the authors knowledge, there are no experimental studies available that show this quantitatively.

In addition, to the best of the authors knowledge, this is the first study to use the ICESat-2 observations to obtain frequency-dependent attenuation rates, to obtain attenuation rates across large spatial scales and across the entire Antarctic, and to observe seasonality of the estimates attenuation rates.

 In summary, even though it is novel to derive attenuation rates from IS2, this paper has methodological flaws that leave questions about the accuracy and validity of the estimate, as discussed in other publications already (Hell and Horvat, 2024). If this paper wants to describe a new

method for calculating attenuation rates, the stated concern give reason why this method is problematic; if the paper wants to derive actual attenuation rates for later use, it needs to qualify its sampling and give reason why this estimate is robust.

additional remarks

I would reword the statement in L.6 that this samples "over a wide range of sea ice

condition".

We respectfully disagree with the reviewer. As this study looks into more than 300 cloud-free tracks, and capturing sea ice conditions across the MIZ, we are confident that the dataset captures a wide range of sea ice conditions.

L 161: "completely different attenuation rates" - that is strong wording, I would remove that.

On the suggestion of the reviewer we will remove 'completely' from this sentence.

L 244: How do we see wave-current interaction in figure 4? I don't follow

We will rephrase the corresponding sentence to clarify.

Fig.8: the coloring choice is unfortunate.

Based on the reviewer's suggestion, the colour of lines will be changed.

---

## Author Comment (AC2)

Reviewer 1

In their manuscript „Finely-resolved along-track wave attenuation…", Joey Voermans and coauthors perform an analysis of the apparent wave attenuation in the Antarctic MIZ based on satellite (ICESat-2) derived along-track profiles of wave energy $E(x)$ for four selected wave frequency bands (corresponding to wave periods of 9, 12, 15 and 18 s). From the original data, published earlier in Brouwer et al., 2022, the Authors extract 320 high-quality $E(x)$ profiles and, assuming exponential attenuation, for each of them compute the corresponding profile of the attenuation coefficient α$(x)$. Based on that data, several statistics of α$(x)$ are computed, including the average $\bar{\alpha}(x)$ for all profiles and for individual months, as well as relationships between α$(x)$ and ice thickness, concentration and $x/x$MIZ (i.e., the distance from the ice edge scaled with MIZ width). An important part of the analysis is related to the influence of spatial variability of sea ice and wave conditions, and of non-zero angles between the satellite tracks and the wave propagation directions on the observed shapes of $E(x)$.

1.      My major comment on the manuscript is this: I fully agree with the Authors that they have at their disposal "a unique dataset of waves in sea ice obtained across a diverse range of Antarctic sea ice conditions" (lines 222-223). Going from a few (in many cases: only two) data points to high-resolution profiles of wave energy is a big step forward. The new dataset makes it possible to find new patterns in the data, but also to verify assumptions that have been commonly used so far in analyses of wave attenuation in sea ice. However, to fully use the potential offered by the new data, new approaches are necessary – whereas my overall impression when reading the manuscript was that the Authors tried their best to make their study methodologically as close to the previous ones as possible, and to keep all old assumptions untouched. Neither the individual $E(x)$ profiles nor the final, average attenuation $\bar{\alpha}$ are exponential. In particular, none of the $E(x)$ curves presented in the figures resembles an exponential curve; in some cases, like e.g. in Fig.1a, the $E(x)$ profile is concave rather than convex, so that the exponential approximation is particularly inadequate. Still, the Authors decide to "use of the commonly adopted assumption that wave energy decays exponentially" (line 105): their analysis begins with piecewise approximation of $E(x)$ with exponential functions. First, as just said, even a quick look at the data is enough to say that this is a poor choice, and second, given the very high spatial resolution of the profiles, no a priori assumptions regarding their shapes are necessary. To the contrary, the shapes can be found as a result of the analysis. Several interesting questions could be answered this way. For instance, is there a one type of function that provides a satisfactory fit to the majority (or a large subset) of the analyzed profiles? Is the fit generally better in the inner MIZ than close to the ice edge? Which function provides a good fit to the average energy profile? Is it really exp(-bx2), as the analysis based on exponential attenuation suggests? If yes, how does this function perform for individual profiles? In previous studies, assumptions regarding the shape of E(x) were necessary given the large spacing between data points, and the choice of exponential shape seemed natural given what we know from theory about attenuation related to individual physical processes. However, the Authors themselves convincingly show in Section 3.2 and Fig. 4 (and 3 as well) that due to many different factors the apparent attenuation observed along "random" satellite tracks, and attenuation that would be observed along wave trains undergoing a certain physical process, are two very different things. In short, in my opinion the analysis can be made more convincing and more valuable for future applications if the Authors rethink their approach to the data and modify the manuscript accordingly.

The reviewer highlights an important question in this field that is yet to be confirmed, i.e., does wave energy truly decay exponentially with distance into the MIZ?
Despite its high spatial resolution, it is not straightforward to answer this question with the given dataset despite its high spatial resolution. The attenuation rate α is strongly dependent on the sea ice conditions. This implies that when sea ice conditions along the transect change, $\alpha$ changes along

the transect as well. If one were to measure the wave energy along a transect it may therefore not necessarily show any sign of exponential decay even if wave energy would truly decay locally as $\exp(-\alpha x)$.

Considering that the wave energy $E(f,x)$ in our study was estimated based on a section length of about 8 km, and that multiple independent estimate of $E(f,x,)$ along a transect are required to confirm whether wave energy really decays exponentially with $x$, sea ice conditions need to remain constant over multiples of 8 km before such an assumption can be reliably tested.

While little is known about the sea ice conditions along each transect, it is unlikely that sea ice conditions are constant at scales of say 30-50 km, and thus, in our opinion, it is most logical to assume piecewise local exponential decay of wave energy in our calculations.

2. I have doubts regarding the widths of the MIZ used in the analysis. The Authors say they used *x*MIZ estimates from Brouwer et al (2022), which are "based on the depth of wave penetration into the MIZ" (line 85), but Fig. 2 and, especially, 3 suggest that these estimates might be inadequate for the present purpose. In Fig. 3b, a reasonable estimate of xMIZ seems to be between 0.6 and 0.7 of the actually used one – which makes a huge difference. At first, I thought that maybe this is just an unfortunately selected example, but the drop in the amount of valid data points in the inner MIZ (red line in Fig.5a) suggests that *x*MIZ used in the analysis is systematically overestimated. Wouldn't it be more reasonable and consistent to compute *x*MIZ(*f*) as the largest *x* for which *E* can be reliably estimated (i.e., for non-grey parts of Fig. 2b and 3b), and only then take a median of those values? The proper estimation of *x*MIZ is very important, because the whole analysis is performed in terms of *x/x*MIZ. In its present form, considering what is presented in Figs. 2, 3 and 5, it is not very convincing.

The definition of the MIZ width is indeed a topic of ongoing research. The frequency dependence of the wave attenuation rate means that short waves dampen faster than long waves, and a definition of the MIZ on the wave energy decay within different frequency bands is therefore problematic. In this study, we look closely at the wave energy contained in narrow frequency ranges rather than the total wave energy. The cutoff threshold of the wave energy presented in our Figure 3 and 4 (gray lines) should not be interpreted as an estimate of the MIZ width, as it strongly depends on the frequency bandwidth used to estimate the wave energy. In Brouwer et al. the frequency dependence is somewhat eliminated by looking at the total wave energy instead: "*The inner boundary of wave penetration is defined as the location where significant wave height attenuation equals the estimated error in significant wave height.*", see Brouwer et al., 2022 (page 2327).

The choice of the definition of Brouwer et al., 2022 is obviously debatable considering the lack of scientific consensus surrounding a definition. A definition based on the threshold of what a method can reliably measure seems, however, unwanted because the MIZ width would change based on the measurement method used (an instrument with higher accuracy would measure a larger MIZ width). However, the definition of Brouwer et al., 2022, could provide a more robust interpretation/derivation of the MIZ width, as this definition seems to be very close to where the wave dominated surface elevation transitions into an ice-structure dominated surface elevation (see Brouwer et al., 2022). Such a definition relies less on the accuracy of the measurement method involved.

3. One of the main results of this analysis is the average (over all profiles) attenuation in function of x/*x*MIZ: it is found that, if exponential attenuation is assumed, it increases linearly with x/*x*MIZ. This result is interesting, but it is hard to think how it could be used in practice (as the Authors suggest in the discussion). First, the relationship α(*x/x*MIZ) is useful only if *x*MIZ is known – which is possible only if we already know the waves! Notably, replacing the wave-based definition of the MIZ with another one (e.g., ice concentration based one) would make the whole approach inconsistent, so it doesn't provide a solution to this problem (there are studies that show that MIZ width estimates based on different criteria can be very different). Second,

this approach seems reasonably simple only in 1D. In 2D, given that the outer and inner boundaries of the MIZ are irregular curves evolving in time, using the scaled coordinate perpendicular to those boundaries would require solving Laplace's equation (as in Strong et al., JPO, 2017) at each model time step. An approach that is anything but simple when one thinks about the details. I think therefore that the statement "it may provide alternative approaches to model wave attenuation in global models if the MIZ width is a known variable" is very misleading.

We agree with the reviewer that this is more challenging than we anticipated. We will therefore add a note to highlight this problem.

4.  I have a question regarding the data in Fig. 5a,b,c. In panels b and c, there are hardly any values of α larger than, say, 5 10-5 m-1. In panel a, values above 5 10-5 m-1 are common in the inner MIZ (*x*>0.7*x*MIZ). Are there no ice thickness and concentration data from those regions, so that they don't appear in b and c? Or what is the reason for the different ranges of α in different plots? The increase of attenuation with ice concentration would be much steeper if all data were included, wouldn't it?

We can confirm that the same data is included in all these figures 5a and 5b. This is not the case for figure 5c as the maximum ice thickness based on SMOS is 50 cm.

The wave attenuation data appears less sorted with sea ice concentration than with the distance into the MIZ because they are more uniformly distributed. For example, see a replotted Fig. 5b below. We note that the gray shaded area in Fig. 5 corresponds to the 25th and 75th percentiles, meaning that 50% of the datapoints are outside this area. In general, one would expect a much stronger correlation with sea ice concentration. This could therefore mean that either this correlation is not as strong in practice, or the data resolution of the ASMR-2 observations is insufficient. We note that based on the study of Montiel et al., 2022, it seems that there is no straightforward scaling of $\alpha$ with sea ice concentration.

[Figure]

5.  In spectral wave models, *cgdE/dx ~ AS*ice + *S*ee (*c*g - group velocity, *A* – ice concentration; 'ee' – everything else). If we forget 'everything else' for a moment, a measure of *S*ice can be obtained by dividing attenuation with ice concentration. I wonder how those profiles would look like. How

much of the observed increase of *dE/dx* with distance from the ice edge can be attributed to the increase in *A*? Is there a zone of high *Sice* close to the ice edge, as suggested by some studies? (A note related to point 1 above: even if we expect that *Sice* represents a single process leading to exponential attenuation, *dE/dx* should be "corrected" for ice concentration before it can be treated as an estimate of *Sice*.)

This is an interesting suggestion of the reviewer and based on this comment we have plotted $\alpha/C$ against $x/x_{MIZ}$ for $T = 12$ s, see below. We may still observe an increase of $\alpha/C$, which suggest that there are also other variables are involved in the damping of wave energy with distance into the MIZ, most likely, sea ice thickness and floe size distribution. We note that this is consistent with our reasoning in the manuscript of the increase in $\alpha$ with distance into the MIZ.
We note that interpretation of $\alpha/C$ near the ice edge is unfortunately not possible due to the limited number of observational points present here. This is because estimates of both $E(f, x)$ and $\alpha$ are taken over a finite distance along the transect, and thus no observations of either can be obtained exactly at the ice edge.

---

## Referee Report (RR1)

This is the review of the revised version of the manuscript „Finely-resolved along-track wave attenuation…" by Joey Voermans and coauthors.

I'll start with a technical issue:
Unfortunately, the version of the manuscript with tracked changes has only the major additions marked. Small corrections and fragments that have been removed are unmarked, which makes the assessment of changes that have been made to the text quite difficult. An advice to the Authors for the future: please use the "trackchanges" or a similar LaTeX package, so that all changes made to the text are clearly visible.

In general, I find that the changes the Authors have made to the manuscript have significantly improved its quality. Most comments and suggestions of the reviewers have been convincingly addressed, and missing details regarding the data analysis added.
However, a few issues remain that, in my opinion, should be solved before the manuscript is suitable for publication in TC. They are related to my previous major comment #1 (notably, the most important one):
The Authors seem to misunderstand the core of that comment – or they just pretend they do in order to avoid the necessity to modify/expand their analysis…
Of course, the sea ice properties do vary at the spatial scales < 8 km (the resolution of the analyzed data). Obviously, they also vary at larger spatial scales, resolved by the analyzed dataset – and, as the results presented in the manuscript clearly show, the net result of that variability are non-exponential profiles of wave energy.
Do the Authors seriously claim that the apparent attenuation – that is, the observed attenuation resulting from the sum of all factors influencing it – miraculously changes character from exponential to non-exponential one when it is observed at spatial scales below and above 8 km?
This is exactly what the Authors assume.
Very importantly: our discussion here is not about attenuation related to any given physical process. In both cases, at the scale of 8 km and at the larger scales resolved by the data, it is the apparent attenuation that is analyzed.
Why should it change character? Why at 8 km? If the resolution of the data equaled, say, 20 or 50 km, would the Authors compute the attenuation in the same way and use the same arguments to justify it?
It's all simply inconsistent and therefore very unconvincing.
In short, I don't find the answer of the Authors to my comment #1 satisfactory.

---

## Referee Report (RR2)

I have been asked to review this manuscript after previous revision iterations already have been taken. The editor asked me, in particular, to assess if and how well the reviewers' comments have been answered.

In my opinion, the authors have provided convincing answers to the points raised by the reviewers and have implemented good updates into the manuscript. At this point, I have very little comments about the manuscript content per se, and I believe that the manuscript presents a good quality analysis of the data investigated, and convincing and best practice methodology. Naturally, there are always limits with accuracy and resolution in time and space with satellite data, but this is already well discussed, and the authors perform a number of mitigation techniques such as averaging and considering statistics.

Therefore, I do not have any major technical comments at this point.

I can recommend a couple of points that could be added in a few sentences to provide a couple more high level discussion points, but these are possibly quite subjective, so these are only recommendations and I understand if the authors do not want to add them

- This paper is an important contribution as it helps take in use satellite data for better understanding waves in ice and, as a consequence, greatly increases the amount of data and the width of the underlying sampling compared with previous small scale analysis. I believe this is a very important aspect of advancing this field and correcting previous mistakes. For example, small sample effects combined with peculiar ice conditions have to the best of my understanding, been proved to be the cause for the wrong equation 1 in the manuscript https://www.nature.com/articles/nature13262 (Storm-induced sea-ice breakup and the implications for ice extent, 2014), which has been "debunked" in a follow up paper by a group of authors including some of the initial authors: "These results suggest that the conclusion in Kohout and others (2014), that large waves decay linearly, is an artefact of analysing a small dataset in different ice conditions." https://doi.org/10.1017/aog.2020.36 . The present work, by allowing to take in use larger dataset, is an important step towards "robustifying" the results presented about waves in ice against such issues.

- A somehow similar issue has been observed regarding the rollover effect, which is now believed to come from noise present specifically in buoys measurements combined with small dataset, as highlighted in "Spurious Rollover of Wave Attenuation Rates in Sea Ice Caused by Noise in Field Measurements" (2021).

These two topics (there may be more similar examples that I am not aware of) have caused significant confusion in our community over the years and both are the consequence of closely related issues: low volume and limited quality of field data. In my view, this is typically the kind of challenges that the present study helps mitigate and correct, which is an important contribution to our community.

---

## Author Response (AR3)

I have been asked to review this manuscript after previous revision iterations already have been taken. The editor asked me, in particular, to assess if and how well the reviewers' comments have been answered.

In my opinion, the authors have provided convincing answers to the points raised by the reviewers and have implemented good updates into the manuscript. At this point, I have very little comments about the manuscript content per se, and I believe that the manuscript presents a good quality analysis of the data investigated, and convincing and best practice methodology. Naturally, there are always limits with accuracy and resolution in time and space with satellite data, but this is already well discussed, and the authors perform a number of mitigation techniques such as averaging and considering statistics.

Therefore, I do not have any major technical comments at this point.

I can recommend a couple of points that could be added in a few sentences to provide a couple more high level discussion points, but these are possibly quite subjective, so these are only recommendations and I understand if the authors do not want to add them

- This paper is an important contribution as it helps take in use satellite data for better understanding waves in ice and, as a consequence, greatly increases the amount of data and the width of the underlying sampling compared with previous small scale analysis. I believe this is a very important aspect of advancing this field and correcting previous mistakes. For example, small sample effects combined with peculiar ice conditions have to the best of my understanding, been proved to be the cause for the wrong equation 1 in the manuscript https://www.nature.com/articles/nature13262 (Storm-induced sea-ice breakup and the implications for ice extent, 2014), which has been "debunked" in a follow up paper by a group of authors including some of the initial authors: "These results suggest that the conclusion in Kohout and others (2014), that large waves decay linearly, is an artefact of analysing a small dataset in different ice conditions." https://doi.org/10.1017/aog.2020.36 . The present work, by allowing to take in use larger dataset, is an important step towards "robustifying" the results presented about waves in ice against such issues.

We agree with the reviewer that large datasets, particularly those that capture a wide range of environmental conditions, are critical in assessing wave dynamics in the polar regions. In particular, small datasets are more likely to be dominated by specific conditions or events (usually because they are obtained within the same region), which complicates the interpretation of any results that follow.

We have therefore added the following in the Discussion & Conclusions section (Lines 299-303 of revised manuscript):

"*Nevertheless, averaging observations across a broad range of sea ice conditions has the advantage that it reduces the likelihood of bias from small sample size effects (e.g., Kohout et al., 2020), where the data may otherwise be disproportionately influenced by environmental conditions that are over-represented in a dataset. As such, the approach followed in our study may provide a more reliable methodology for identifying the overall effects of sea ice properties on wave attenuation rates.*"

- A somehow similar issue has been observed regarding the rollover effect, which is now believed to come from noise present specifically in buoys measurements combined with small dataset, as highlighted in "Spurious Rollover of Wave Attenuation Rates in Sea Ice Caused by Noise in Field Measurements" (2021).

We agree with the reviewer that this is a critical quality control consideration. The spurious rollover effect is not necessarily caused by small datasets, but rather by not considering an appropriate signal-to-noise ratio threshold during data quality control. This, if not appropriately addressed, will lead (and has led) to erroneous results/observations/conclusions. We note, however, that this was briefly mentioned already in the Results section (Line 160), and we therefore decided not to discuss this further in the Discussion section.

These two topics (there may be more similar examples that I am not aware of) have caused significant confusion in our community over the years and both are the consequence of closely related issues: low volume and limited quality of field data. In my view, this is typically the kind of challenges that the present study helps mitigate and correct, which is an important contribution to our community.